# Structure of human PIEZO1 and its slow-inactivating channelopathy mutants

**Yuanyue Shan[1,2†], Xinyi Guo[1†], Mengmeng Zhang[1†], Meiyu Chen[1†], Ying Li[1], Mingfeng Zhang[1,2,3], Duanqing Pei[1,2]\***

[1]Laboratory of Cell Fate Control, School of Life Sciences, Westlake University, Hangzhou, China; [2]Westlake Laboratory of Life Sciences and Biomedicine, Hangzhou, Zhejiang, China; [3]Fudan University, Shanghai, China

## eLife Assessment

This is a **useful** revised manuscript that shows a set of data including the first cryo-EM structures of human PIEZO1 as well as structures of disease-related mutants in complex with the regulatory subunit MDFIC, which generate different inactivation phenotypes. The molecular basis of PIEZO channel inactivation is of great interest due to its association with several pathologies. This manuscript provides some structural insights that may help to ultimately build a molecular picture of PIEZO channel inactivation. While the structures are of use and clear conformational differences can be seen in the presence of the auxiliary subunit MDFIC, the strength of the evidence supporting the conclusions of the paper, especially the proposed role for pore lipids in inactivation, is **incomplete**.

**\*For correspondence:**
peiduanqing@westlake.edu.cn

[†]These authors contributed equally to this work

**Competing interest:** The authors declare that no competing interests exist.

**Abstract** PIEZO channels transmit mechanical force signals to cells, allowing them to make critical decisions during development and in pathophysiological conditions. Their fast/slow inactivation modes have been implicated in mechanopathologies but remain poorly understood. Here, we report several near-atomic resolution cryo-EM structures of fast-inactivating wild-type human PIEZO1 (hPIEZO1) and its slow-inactivating channelopathy mutants with or without its auxiliary subunit MDFIC. Our results suggest that hPIEZO1 has a more flattened and extended architecture than curved mouse PIEZO1 (mPIEZO1). The multi-lipidated MDFIC subunits insert laterally into the hPIEZO1 pore module like mPIEZO1, resulting in a more curved and extended state. Interestingly, the high-resolution structures suggest that the pore lipids, which directly seal the central hydrophobic pore, may be involved in the rapid inactivation of hPIEZO1. While the severe hereditary erythrocytosis mutant R2456H significantly slows down the inactivation of hPIEZO1, the hPIEZO1-R2456H-MDFIC complex shows a more curved and contracted structure with an inner helix twist due to the broken link between the pore lipid and R2456H. These results suggest that the pore lipids may be involved in the mechanopathological rapid inactivation mechanism of PIEZO channels.

## Introduction

Cells rely on mechanosensitive (MS) channels, which rapidly convert force into ion flow, to sense environmental changes and make appropriate decisions throughout the prokaryotic and eukaryotic kingdoms. In bacteria, two major MS channels, the small-conductance mechanosensitive channel (MscS) (*Levina et al., 1999*) and large-conductance mechanosensitive channel (MscL) (*Sukharev et al., 1994*), are responsible for bacterial force sensing. In contrast, in mammals, only the PIEZO (*Coste et al., 2010*), two-pore domain K+ channel (K2P) (*Maingret et al., 1999*) and TMEM63 (*Murthy et al., 2018*; *Zhang et al., 2018*), channels have been identified as bona fide MS channels. Structural studies indicate that not all MS channels are constructively conserved, ranging from monomer (*Zhang et al., 2023*)

to heptamer (*Bass et al., 2002*), but they are likely to obey two putative principles of force-from-lipid (FFL) and force-from-filament (*Anishkin et al., 2014*; *Martinac et al., 1990*). Within the heptameric MscS channel in bacteria, three types of lipids, the pore lipids, the gatekeeper lipids, and the pocket lipids, sense and transduce force at corresponding positions (*Zhang et al., 2021*). Similarly, the pseudo-tetrameric MS K2P channels also utilize at least three types of lipids to regulate channel activation (*Brohawn et al., 2014*; *Schmidpeter et al., 2023*). In particular, pore lipids seal the channel pores and could be removed by the mechanical force upon activation of the MscS, MS K2P, and TMEM63 channels (*Zhang et al., 2023*; *Zhang et al., 2021*; *Brohawn et al., 2014*). In the trimeric PIEZO channels, membrane curvature is likely associated with channel activation (*Lin et al., 2019*; *Mulhall et al., 2023*; *Yang et al., 2022*). Although the PIEZO channel also obeys the FFL principle (*Zheng et al., 2019*), the role of the lipids in PIEZO channel gating remains elusive.

PIEZOs from different species or in disparate native cells exhibit diverse inactivation kinetics. Human PIEZO1 (hPIEZO1) exhibits faster inactivation gating kinetics than mouse PIEZO1 (mPIEZO1) under the same voltage condition (*Coste et al., 2010*; *Zheng et al., 2019*; *Bae et al., 2013*), so mPIEZO1 presents a native gain-of-function ('GOF') state relative to hPIEZO1 regarding inactivation gating kinetics. Also, it has been reported recently that the MyoD family inhibitor, MDFIC/MDFI, is an auxiliary subunit of PIEZO channels to prolong inactivation and reduce mechanosensitivity (*Zhou et al., 2023*). Besides, some hereditary erythrocytosis (HX) mutations exhibit slower inactivation gating kinetics than wild-type hPIEZO1. The slower inactivation may result in the higher open probability and delayed inactivation of hPIEZO1; thus, these mutations are considered GOF (*Bae et al., 2013*). HX, also known as inherited dehydrated stomatocytosis, is an autosomal dominant disorder that causes dehydration of red blood cells, resulting in hemolytic anemia. In addition to changes in gating kinetics, some of the HX mutations show alterations in response to osmotic pressure and in membrane protein trafficking (*Glogowska et al., 2017*). Interestingly, the populations with a mild GOF PIEZO1 allele are likely to be resistant to malaria infection, while some present a strong association with increased plasma iron (*Ma et al., 2021*; *Ma et al., 2018*). Although the inactivation phenotype of PIEZO channels is very susceptible to intrinsic and extrinsic factors, the molecular basis of their inactivation remains unclear.

Here, we report the structural basis of hPIEZO1 gating and inactivation based on the architecture of hPIEZO1 and its slow-inactivating channelopathy mutants with or without its auxiliary subunit MDFIC at near-atomic resolution by cryo-EM. Our high-resolution structures support a model in which the pore lipids directly seal the central hydrophobic pore and involve fast inactivation of hPIEZO1. This model also provides a mechanistic understanding of the severe HX mutant R2456H with a more curved and contracted structure with an inner helix (IH) twist due to the broken link between the pore lipid and R2456H.

## Results

### Overall structure of the full-length hPIEZO1

hPIEZO1 is known to have faster inactivation gating kinetics than mPIEZO1 (*Bae et al., 2013*). The structural basis of this difference remains unresolved. To resolve this, we synthesized a codon-optimized full-length hPIEZO1 with a C-terminal GFP and flag tag into the pBM vector (*Goehring et al., 2014*), and this construct exhibits similar mechanosensitivity and fast inactivation gating properties as previously reported by whole-cell poking assay (*Figure 1*, *Figure 1—figure supplement 1*). We overexpressed hPIEZO1 in 12 l of HEK293F suspension cells, extracted the protein with lauryl maltose neopentyl glycol (LMNG), and purified it in a digitonin environment. The symmetric peak of the full-length hPIEZO1 suggests the protein is homogenous. The smeared bands around 300 kDa on SDS-PAGE indicate the potential presence of posttranslational modifications (PTMs), and the lower bands may be the binding partners or degraded fragments of hPIEZO1 (*Figure 1—figure supplement 2a and b*).

The purified hPIEZO1 protein was subjected to the standard cryo-EM workflow, resulting in a cryo-EM density map at a total resolution of 3.3 Å from approximately 87K particles (*Figure 1—figure supplement 3a–e*) that allowed us to build a near-atomic model of hPIEZO1 (Table 1, *Figure 1—figure supplement 4*). Among the nine transmembrane helix units (THUs) containing four transmembrane helices for each predicted by primary sequences, the two N-terminal ones are not visible in the cryo-EM density map. The rest of the THUs, along with the pore module consisting of an anchor, an

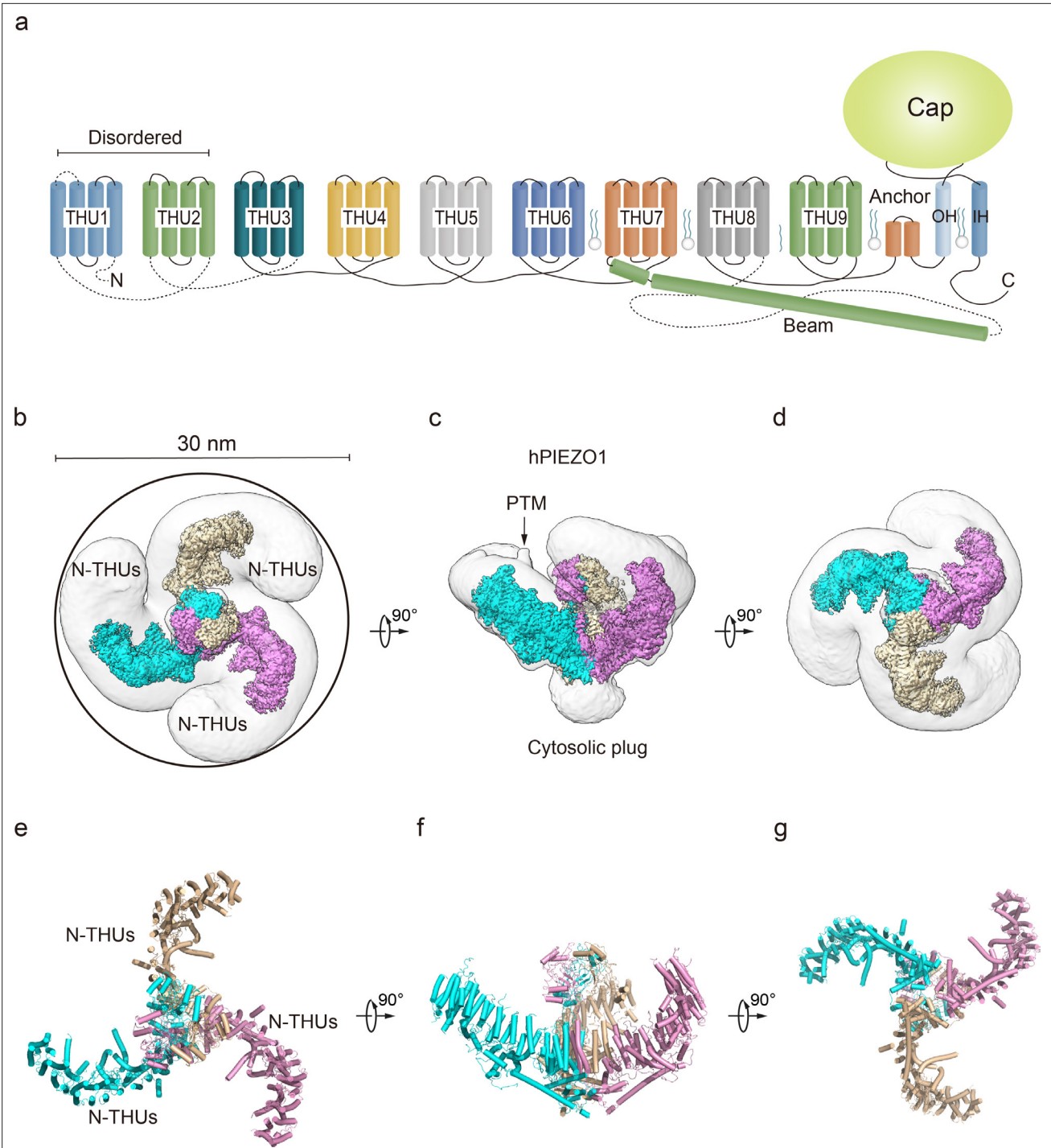

**Figure 1.** Structure of full-length human PIEZO1 (hPIEZO1) channel. (**a**) 38-TM topology model of a single hPIEZO1 subunit. The nine transmembrane helix units (THUs), a long beam helix, an anchor domain, and two pore module helices (OH and IH, respectively). Each THU contains four transmembrane helices. THU1 and THU2 are likely disordered and therefore not visible in the hPIEZO1 cryo-EM density map. (**b**) The 3.3 Å cryo-EM density map of hPIEZO1 at the top view. The density of each single subunit is colored in cyan, wheat, and pink, respectively. Around 30 nm digitonin disk is shown as gray density by low pass filtering. (**c**) The 3.3 Å cryo-EM density map of hPIEZO1 at side view. The density of each single subunit is colored in cyan, wheat, and pink, respectively. The potential posttranslational modification (PTM) at the THUs region is indicated. A flexible density binds to the cytosolic plug, which may stand for an additional hPIEZO1 auxiliary. (**d**) The 3.3 Å cryo-EM density map of hPIEZO1 at the bottom view. The density of each single subunit is colored in cyan, wheat, and pink, respectively. (**e**) The cartoon model of hPIEZO1 at the top view. Each single subunit is colored in cyan, wheat, and pink, respectively. (**f**) The cartoon model of hPIEZO1 at the side view. Each single subunit is colored in cyan, wheat, and pink, respectively. (**g**) The cartoon model of hPIEZO1 at the bottom view. Each single subunit is colored in cyan, wheat, and pink, respectively.

*Figure 1 continued on next page*

*Figure 1 continued*

The online version of this article includes the following source data and figure supplement(s) for figure 1:

**Figure supplement 1.** The whole-cell poking evoked currents of wild-type human PIEZO1 (hPIEZO1), the gain-of-function (GOF) channelopathy mutants with or without MDFIC.

**Figure supplement 1—source data 1.** SDS-PAGE gel sorce data of purifed hPIEZO1 and its channelopathy mutants with or without hMDFIC.

**Figure supplement 2.** Purification procedures of human PIEZO1 (hPIEZO1), the channelopathy mutants with or without MDFIC proteins.

**Figure supplement 3.** Cryo-EM data processing procedure of wild-type (WT) human PIEZO1 (hPIEZO1).

**Figure supplement 4.** The EM density of wild-type (WT) human PIEZO1 (hPIEZO1).

outer helix (OH), an IH, and a cap domain, are resolved (*Figure 1a*). Compared to the anchor, OH, and IH, the resolution of the cap domain is slightly lower, which may be due to the flexibility of the cap domain. On the intracellular side, a long beam helix supports the THU7–9 and pore modules. Each hPIEZO1 subunit has a curved blade structure, and the three subunits form the trimeric PIEZO channel, exhibiting a bowl-like shape. The hPIEZO1 density map, filtered with a low-pass filter, shows a disk approximately 30 nm wide with a horn-like density in the peripheral blade region, likely due to the PTMs such as glycosylation. Interestingly, we see a large density below the central pore module at the intracellular side, which may be potential auxiliary subunits, consistent with the lower molecular weight bands on SDS-PAGE (*Figure 1b–g*, *Figure 1—figure supplement 2b*).

## hPIEZO1 is more flattened and extended than mPIEZO1

Several studies using high-speed atomic force microscopy (*Lin et al., 2019*), cryo-EM (*Yang et al., 2022*), and 3D interferometric photoactivation localization microscopy (*Mulhall et al., 2023*) suggest that PIEZO blades may receive mechanical force through sensing membrane curvature. The reshaping of the curvature of the PIEZO conducts force to the pore region, which may result in the channel pore opening and ion flow. Structural comparison of hPIEZO1 and curved mPIEZO1 shows that hPIEZO1 presents a more flattened structure from the side view (*Figure 2a*). The blades of hPIEZO1 are about 5 Å down toward the cytoplasmic side than those of mPIEZO1. Meanwhile, the cap domain of hPIEZO1 is slightly upward to the extracellular side (*Figure 2a*). In the top view, the distal blades rotate counterclockwise about 22 Å compared to mPIEZO1 (*Figure 2b*). Therefore, hPIEZO1 appears more extended than the curved mPIEZO1. Meanwhile, hPIEZO1 still maintains a sizeable curved state compared to the flattened mPIEZO1 structure. From the top view, the two structures are very similar (*Figure 2c and d*). The pore radius of hPIEZO1 is between that of the curved and flattened mPIEZO1. Therefore, the curved hPIEZO1 may still represent a nonconducting state (*Figure 2e–h*), and the differences in curvature and pore radius may be implicated in channel gating properties of hPIEZO1 and mPIEZO1.

## GOF channelopathy mutants, hPIEZO1-A1988V, hPIEZO1-E756del, and hPIEZO1-R2456H, are structurally unstable

The success in solving the hPIEZO1 structure encouraged us to probe the structural basis of channelopathy. We selected the mild GOF mutants hPIEZO1-A1988V and hPIEZO1-E756del, familiar in African populations with a slightly longer inactivation time, and the more severe R2456H mutant with a much longer inactivation time (*Bae et al., 2013*; *Ma et al., 2018*). However, when we tried to solve their structures based on the same method, i.e., overexpressing and purifying these three channelopathy mutants in HEK293F suspension cells, extracting them with LMNG detergent and purifying them in a digitonin environment (*Figure 1—figure supplement 2c–e*), we can only obtain three channelopathy mutant proteins and their raw cryo-EM images. Still, we failed to achieve satisfactory 2D class averages for them (*Figure 4—figure supplements 1 and 3*, *Figure 5—figure supplement 1*). As a result, we could not generate high-resolution 3D density maps of hPIEZO1-E756del and hPIEZO1-R2456H. We then tried and obtained almost twice as many hPIEZO1-A1988V images as the wild type but only achieved a 3.7 Å resolution cryo-EM density map (*Table 1*, *Figure 4—figure supplement 1*). Yet, the resulting overall structure is similar to the wild type (*Figure 4—figure supplement 2*). Based on these results, these three hPIEZO1 channelopathy GOF mutants may be structurally less stable than wild type.

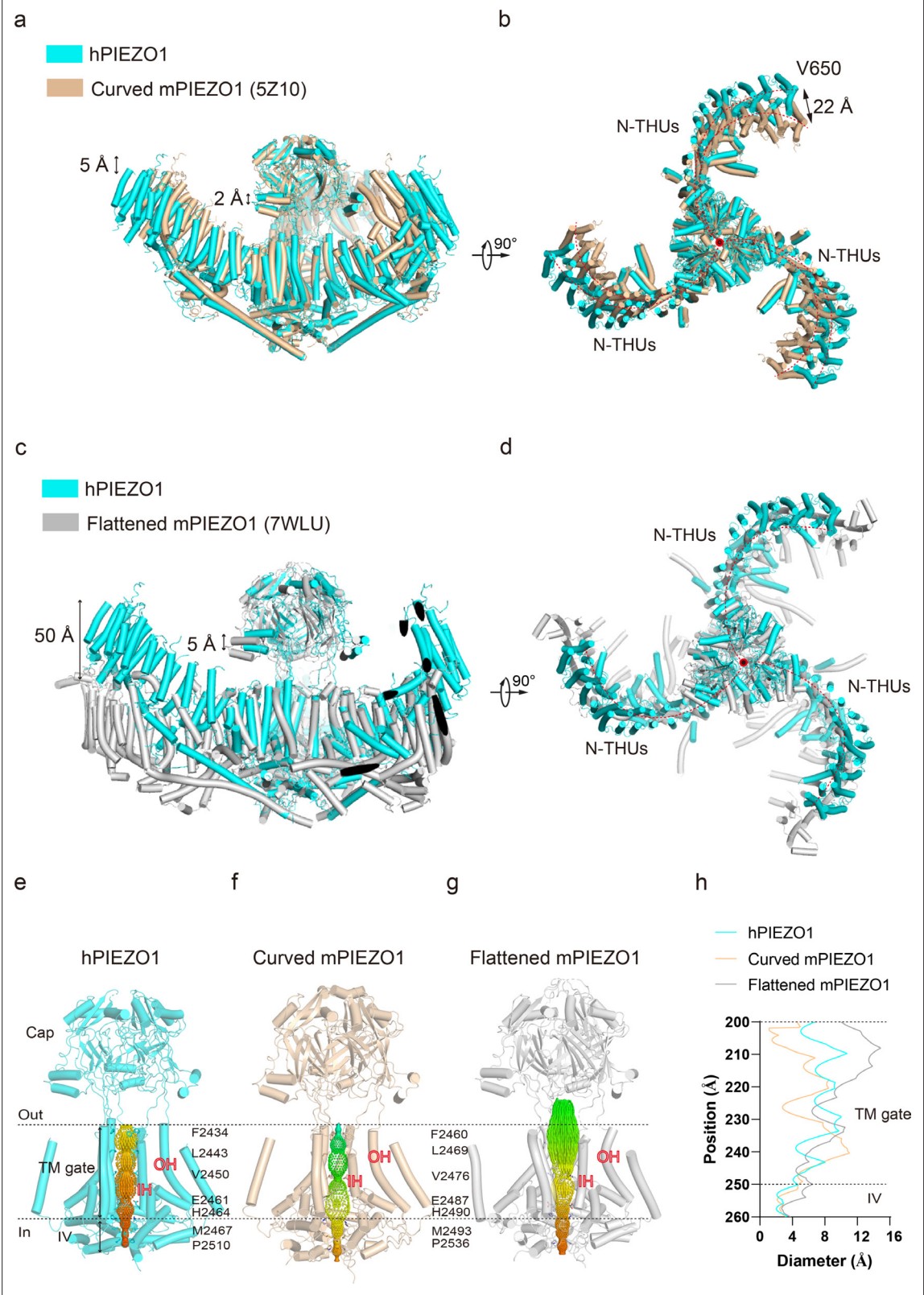

**Figure 2.** Structural comparison of human PIEZO1 (hPIEZO1) and mouse PIEZO1 (mPIEZO1). (**a**) Structural comparison of hPIEZO1 (this study) and curved mPIEZO1 (5Z10) at side view. The distance of the distal blade between curved mPIEZO1 and hPIEZO1 is measured as 5 Å, and the distance of the cap between curved mPIEZO1 and hPIEZO1 is measured as 2 Å. (**b**) Structural comparison of hPIEZO1 (this study) and curved mPIEZO1 (5Z10) at top view. The distance between V650 residue in hPIEZO1 and the corresponding residue in curved mPIEZO1 is around 22 Å; therefore, the blades of

*Figure 2 continued on next page*

*Figure 2 continued*

hPIEZO1 present a contracted state from the top view. The blade center lines are shown as red dashed lines. (**c**) Structural comparison of hPIEZO1 (this study) and flattened mPIEZO1 (7WLU) at side view. Compared to the flattened mPIEZO1, the mimic membrane curvature of hPIEZO1 is between the curved and flattened mPIEZO1. (**d**) Structural comparison of hPIEZO1 (this study) and curved mPIEZO1 (5Z10) at top view. hPIEZO1 presents a similar extended state with the flattened mPIEZO1. (**e**) The cartoon model of the hPIEZO1 pore module with calculated pore. (**f**) The cartoon model of curved mPIEZO1 pore module with calculated pore. (**g**) The cartoon model of flattened mPIEZO1 pore module with calculated pore. (**h**) The calculated pore diameter of hPIEZO1 (cyan), curved mPIEZO1 (wheat), and flattened mPIEZO1 (gray) along the z axis.

### The structure of hPIEZO1-MDFIC complex

MDFIC, a MyoD family inhibitor protein, has recently been identified as an auxiliary subunit of piezo channels capable of decelerating channel inactivation and attenuating channel mechanosensitivity (*Zhou et al., 2023*). To solve the structural basis of MDFIC on hPIEZO1, we co-expressed it with hMDFIC in HEK293F and purified the complex (*Figure 1—figure supplement 2a and f*). We obtained a 3.0 Å resolution cryo-EM density map of the hPIEZO1-MDFIC complex (*Table 1*, *Figure 3—figure supplement 1*). The overall cryo-EM density map is similar to hPIEZO1 alone with a disk of about 30 nm (*Figure 3a–f*), but allowed us to validate the MDFIC density with multi-lipidated cysteines on the C-terminal amphipathic helix (*Figure 3—figure supplement 2*). This C-terminal helix inserted laterally into the pore module where the exact position of mPIEZO1 is found (*Figure 3a–f*). Interestingly, while MDFIC does not significantly alter the curvature of mPIEZO1, it induces a more curved and contracted architecture in hPIEZO1 from the side and top views, respectively (*Figure 3g and h*). Combining with the similar effect of MDFIC on mPIEZO1 (*Zhou et al., 2023*) and hPIEZO1 (*Figure 1—figure supplement 1*), we suspect that MDFIC decelerates hPIEZO1 inactivation and weakens its mechanosensitivity by remodeling the pore module and exerting more blade curvature.

### MDFIC stabilizes GOF channelopathy hPIEZO mutant complexes

The curving effect of MDFIC on hPIEZO1 further encouraged us to test if it can stabilize the unstable channelopathy mutants hPIEZO1-A1988V, hPIEZO1-E756del, and hPIEZO1-R2456H (*Figure 4a*). Indeed, electrophysiological studies showed that co-expression of these channelopathy mutants with MDFIC resulted in significantly reduced mechanosensitivity and inactivation rate (*Figure 1—figure supplement 1b*). We then co-expressed these hPIEZO1 mutants with MDFIC in HEK293F suspension cells and purified the resulting complex using the same method (*Figure 1—figure supplement 2g–i*). Surprisingly, with standard cryo-EM analysis, we obtained high-quality 2D class averages for hPIEZO1-A1988V-MDFIC and hPIEZO1-E756del-MDFIC. Cryo-EM density maps were obtained for hPIEZO1-A1988V-MDFIC and hPIEZO1-E756del-MDFIC with overall resolutions of 3.1 and 2.8 Å, respectively (*Table 1*, *Figure 4—figure supplements 4 and 5*). The densities of the multi-lipidated C-terminal amphipathic helix of MDFIC are clearly present in both the hPIEZO1-A1988V-MDFIC and hPIEZO1-E756del-MDFIC maps. For hPIEZO1-R2456H-MDFIC, the 2D class averages show a clear trimer state (*Figure 5—figure supplement 2b*). The resolution of hPIEZO1-R2456H-MDFIC was improved to 4.6 Å with 16K particles, allowing us to build a relatively accurate model of the transmembrane helices based on the hPIEZO1-MDFIC model (*Table 1*, *Figure 5—figure supplements 2 and 3*). These results further suggest that MDFIC stabilizes the GOF channelopathy mutants structurally.

### hPIEZO1-R2456H-MDFIC with more curved blades but more extended pore

The unexpected higher-resolution maps of hPIEZO1-A1988V-MDFIC and hPIEZO1-E756del-MDFIC reveal a bowl-like disk of about 30 nm for both complexes (*Figure 4h–j*). On the other hand, the hPIEZO1-R2456H-MDFIC shows a smaller disk of 25 nm compared to the 30 nm for the wild type and the other two mutants (*Figure 5a–f*). The smaller disk is mainly due to the more significant contraction of the blade arms and a more curved state compared to the wild-type hPIEZO1-MDFIC in the side view (*Figure 5g and h*). Surprisingly, the IH of hPIEZO1-R2456H-MDFIC showed a significant twist of ~35° compared to wild-type hPIEZO1-MDFIC (*Figure 6g and h*). The coiled-coil shape of the IH pore results in a more dilated pore on the extracellular side (*Figure 5i–l*), suggesting that unlike the mild GOF E756del and A1988V mutations present in the typical African population and located in the THUs region, the more severe mutation R2456H, which is located in the IH, not only leads to blade reshaping but also results in pore expansion (*Figure 6d–h*).

**Table 1.** Cryo-EM data collection, refinement, and validation statistics.

| | hPIEZO1 | hPIEZO1-A1988V | hPIEZO1 Co MDFIC | hPIEZO1-A1988V CoMDFIC | hPIEZO1-E756del Co MDFIC | hPIEZO1-R2456H Co MDFIC |
|---|---|---|---|---|---|---|
| **Data collection and processing** | | | | | | |
| Microscope | FEI Titan Krios | FEI Titan Krios | FEI Titan Krios | FEI Titan Krios | FEI Titan Krios | FEI Titan Krios |
| Magnification | 105000 | 215000 | 215000 | 215000 | 215000 | 215000 |
| Voltage (KV) | 300 | 300 | 300 | 300 | 300 | 300 |
| Detector | Gatan K3 | Falcon 4i | Falcon 4i | Falcon 4i | Falcon 4i | Falcon 4i |
| Electron exposure (e-/ Å2) | 60 | 60 | 40 | 40 | 40 | 40 |
| Defocus range (µm) | –0.9 to –1.3 | –0.9 to –1.3 | –0.9 to –1.3 | –0.9 to –1.3 | –0.9 to –1.3 | –0.9 to –1.3 |
| Pixel size (Å) | 0.849 | 0.849 | 0.57 | 0.57 | 0.57 | 0.57 |
| Symmetry imposed | C3 | C3 | C3 | C3 | C3 | C3 |
| Initial particle images (no.) | 264225 | 87296 | 105921 | 114028 | 336158 | 139340 |
| Final particle images (no.) | 87369 | 37188 | 46605 | 41164 | 89754 | 15975 |
| Map resolution (Å) | 3.3 | 3.75 | 3.07 | 3.16 | 2.83 | 4.52 |
| FSC threshold | 0.143 | 0.143 | 0.143 | 0.143 | 0.143 | 0.143 |
| **Refinement** | | | | | | |
| Initial model used (PDB code) | 7WLT | 7WLT | 7WLT | 7WLT | 7WLT | 7WLT |
| Model resolution (Å) | 3.3 | 3.8 | 3 | 3.2 | 2.8 | 4.6 |
| FSC threshold | 0.143 | 0.143 | 0.143 | 0.143 | 0.143 | 0.143 |
| Map sharpening B factor (Å2) | 79 | 96 | 52 | 52 | 63 | 108 |
| **Model composition** | | | | | | |
| Non-hydrogen atoms | 31695 | 31695 | 32145 | 32145 | 32145 | 32124 |
| Protein residues | 3852 | 3852 | 3915 | 3915 | 3915 | 3915 |
| Water | 0 | 0 | 0 | 0 | 0 | 0 |
| Ions | 0 | 0 | 0 | 0 | 0 | 0 |
| **B factors** | | | | | | |
| Protein | 182.42 | 192.52 | 144.47 | 142.11 | 143.26 | 717.62 |
| Ligand | / | / | / | / | / | / |
| Water | / | / | / | / | / | / |
| **R.m.s. deviations** | | | | | | |
| Bond lengths (Å) | 0.012 | 0.013 | 0.011 | 0.011 | 0.011 | 0.017 |
| Bond angles (°) | 1.172 | 1.184 | 1.139 | 1.137 | 1.13 | 2.002 |
| **Valiation** | | | | | | |
| MolProbity score | 2.21 | 2.23 | 2.2 | 2.2 | 2.2 | 2.92 |
| Clash score | 15.69 | 16.01 | 15.98 | 15.38 | 15.92 | 59 |
| Poor rotamers (%) | 0.03 | 0.03 | 0.03 | 0.03 | 0.03 | 0 |
| **Ramachandran plot** | | | | | | |

*Table 1 continued on next page*

*Table 1 continued*

| | hPIEZO1 | hPIEZO1-A1988V | hPIEZO1 Co MDFIC | hPIEZO1-A1988V CoMDFIC | hPIEZO1-E756del Co MDFIC | hPIEZO1-R2456H Co MDFIC |
|---|---|---|---|---|---|---|
| Favored (%) | 91.19 | 90.34 | 91.77 | 91.95 | 91.94 | 84.19 |
| Allowed (%) | 8.07 | 8.32 | 7.66 | 7.62 | 7.54 | 14.35 |
| Disallowed (%) | 0.75 | 1.34 | 0.58 | 0.53 | 0.52 | 1.47 |

## R2456 anchors a lipid at the pore module

Then, we wished to build a pore module based on the solved structures. We were surprised to find an apparent lipid density in wild-type hPIEZO1-MDFIC, hPIEZO1-A1988V-MDFIC, and even more evident in the 2.8 Å resolution map of hPIEZO1-E756del-MDFIC in the pore module, where one of its hydrophobic fatty acid tails inserts into the hydrophobic pore formed by I2447, V2450, and F2454, thus sealing the pore to prevent ion flow (*Figure 6a–c*). Even more surprisingly, the hydrophilic phosphate head interacts directly with R2456, a residue with the more severe form of GOF mutation, on the side of the IH of the pore. In addition, another fatty acid chain of the pore lipid interacts with the acyl chains of the covalently linked MDFIC lipids, forming a stable hPIEZO1-multi-lipidated MDFIC-pore lipid complex (*Figure 6a–c*). We also carefully checked the corresponding pore lipid configuration in the 3.3 Å resolution map of hPIEZO1 and the 3.8 Å resolution map of hPIEZO1-A1988V without the multi-lipidated MDFIC auxiliary subunit and found that in wild-type hPIEZO1, similar pore lipids also insert into the hydrophobic pore interacting with R2456 (*Figure 7a and f*). However, in the mild GOF hPIEZO1-A1988V, which has a slower inactivation rate compared to wild-type hPIEZO1, the acyl chains of the pore lipid are retracted from the central hydrophobic pore to the side of the IH, although the hydrophilic phosphate group heads still interact with R2456 (*Figure 7b and f*). These results reveal a critical role of R2456 in anchoring lipids at the pore module.

## Pore lipids may be involved in the fast inactivation of hPIEZO1

Our results support a putative model that the hydrophobic acyl chain tails of the pore lipids insert into the hydrophobic pore region and seal the pore (*Figure 8a*). Consistently, the slower inactivating hPIEZO1-A1988V mutant has the same hydrophobic acyl chain tails retracted from the hydrophobic pore region, implying that the pore lipids are involved in the fast inactivation of hPIEZO1 (*Figure 8a and c*). The evidence supporting this model is based on previous electrophysiological functional studies that substitution of the hydrophobic pore, formed by I2447, V2450, and F2454, with a hydrophilic pore prolongs the inactivation time for both PIEZO1 and PIEZO2 channels (*Zheng et al., 2019*). More robust evidence comes from the HX channelopathy mutant R2456H, wherein the interaction between the hydrophilic phosphate group head and R2456 is disrupted, which leads to remodeling of the blade and pore module (*Figure 6d–h*), thus significantly prolonging the inactivation time. These results suggest that the pore lipids are involved in the fast inactivation of hPIEZO1 (*Figure 8d*). Consistently, in curved and flattened mPIEZO1 structures (*Yang et al., 2022*), pore lipids are occupying the similar lateral side of the pore, but not sealing the pore like the hPIEZO1-A1988V mutant (*Figures 7d–e and 8c*). Similarly, we suspect that the curved and flattened mPIEZO1 structures are not sealed by pore lipid and thus may not adopt the deep inactivation state, consistent with electrophysiological results showing a slower inactivation manner of mPIEZO1 (*Coste et al., 2010*; *Bae et al., 2013*).

The multi-lipidated MDFIC functions to stabilize the pore lipids that seal the hydrophobic pore further through interactions between hydrophobic acyl chain tails of the pore lipids and the lipids covalently linked to MDFIC (*Figure 7c and f*). Once activated, the PIEZO-MDFIC complexes assume much slower inactivation (*Figure 1—figure supplement 1*), consistent with the model that the multi-lipidated MDFIC makes the PIEZO challenging to open by mechanical force (*Figure 8b and d*). Therefore, we deduce that the hPIEZO1-MDFIC/hPIEZO1-A1988V-MDFIC/hPIEZO1-E756del-MDFIC reshape the pore module and fall into a deep resting state, in which it becomes rather tricky to remove the pore lipid from the hydrophobic pore by mild mechanical force. On the other hand, once the pore lipids are removed by a higher threshold of the mechanical force, it is equally more difficult for the pore lipids to return to the hydrophobic pore region and regain the stable pore

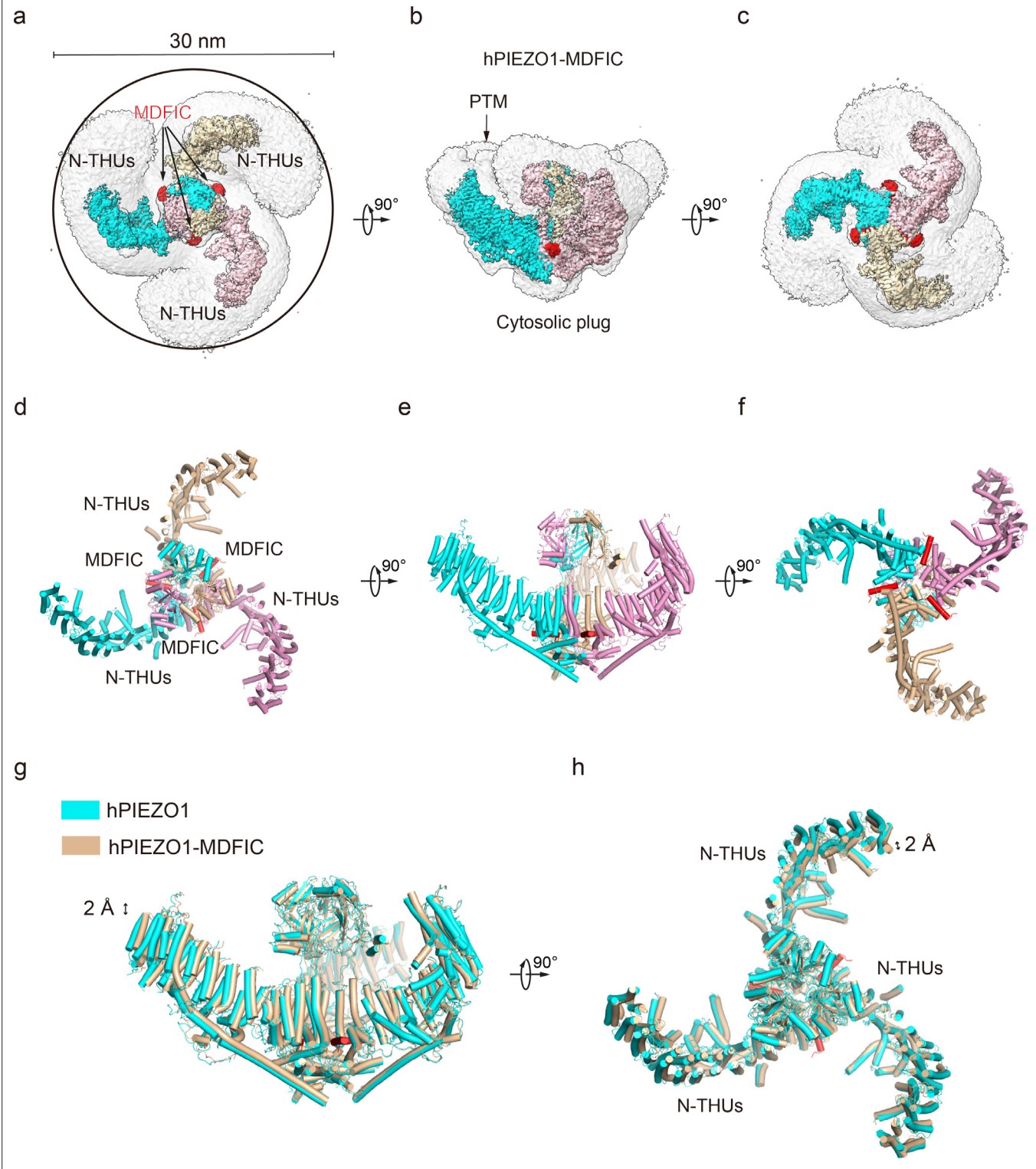

**Figure 3.** Structure of full-length human PIEZO1 (hPIEZO1)-MDFIC complex. (**a**) The 3.0 Å cryo-EM density map of hPIEZO1-MDFIC at top view. The density of each single subunit is colored in cyan, wheat, and pink, respectively. The density of the three C-terminal helices of MDFIC is colored in red. Around 30 nm digitonin disk is shown as gray density by low pass filtering. (**b**) The 3.0 Å cryo-EM density map of hPIEZO1-MDFIC at the side view. The density of each single subunit is colored in cyan, wheat, and pink, respectively. The density of the three C-terminal helices of MDFIC is colored in red. The potential posttranslational modification (PTM) at the transmembrane helix units (THUs) region is indicated. A flexible density binds to the cytosolic plug, which may stand for an additional hPIEZO1 auxiliary subunit. (**c**) The 3.0 Å cryo-EM density map of hPIEZO1-MDFIC at bottom view. The density of

*Figure 3 continued on next page*

*Figure 3 continued*

each single subunit is colored in cyan, wheat, and pink, respectively. The density of three C-terminal helices of MDFIC is colored in red. (**d**) The cartoon model of hPIEZO1-MDFIC at the top view. Each single subunit is colored in cyan, wheat, and pink, respectively. Three C-terminal helices of MDFIC are colored in red. (**e**) The cartoon model of hPIEZO1-MDFIC at the side view. Each single subunit is colored in cyan, wheat, and pink, respectively. Three C-terminal helices of MDFIC are colored in red. (**f**) The cartoon model of hPIEZO1-MDFIC at the bottom view. Each single subunit is colored in cyan, wheat, and pink, respectively. Three C-terminal helices of MDFIC are colored in red. (**g**) Structural comparison of hPIEZO1 and hPIEZO1-MDFIC at side view. The motion of the distal blade between hPIEZO1 and hPIEZO1-MDFIC is around 2 Å from the side view. (**h**) Structural comparison of hPIEZO1 and hPIEZO1-MDFIC at top view. The motion of the distal blade between hPIEZO1 and hPIEZO1-MDFIC is around 2 Å from the top view.

The online version of this article includes the following figure supplement(s) for figure 3:

**Figure supplement 1.** Cryo-EM data processing procedure of human PIEZO1 (hPIEZO1) co MDFIC.

**Figure supplement 2.** The EM density of human PIEZO1 (hPIEZO1)-E756del-MDFIC.

module-multi-lipidated MDFIC-pore lipid complex, thus resulting in the very prolonged slow inactivation phenotype (*Figure 8b and d*).

## Discussion

Piezo ion channels are the critical force sensors (*Coste et al., 2010*) that allow cells to sense their physical environment and regulate cell fate. Intracellular signaling pathways have become the focus of research to understand the mechanism of mechanical force involved by piezo channels in cells (*Song et al., 2019*). MyoD was the first transcription factor identified to specify cell fate in a cell-autonomous fashion, ushering decades of investigation into cell fate control that led to the discovery of iPSC reprogramming (*Chambers and Studer, 2011*). It is intriguing that MyoD interacting proteins MDFIC and MDFI are auxiliary factors for the PIEZO MS ion channels, suggesting that these complexes link cell force sensing and cell fate control (*Zhou et al., 2023*). Indeed, mice with PIEZO deletions are lethal, supporting the idea that mechanosensing through these channels is critical in cell fate control during development. To understand the role of PIEZO or PIEZO-MDFIC/MDFI in human cell fate control, we solved the structure of hPIEZO1 in complex with and without MDFIC. MDFIC enables hPIEZO1 to respond to different forces by modifying the pore module through lipid interactions.

The fast and slow inactivation modes may allow cells to respond to external forces more accurately. Indeed, inactivation is widespread in different types of ion channels. Inactivation can be plastic, driven by intrinsic and extrinsic cues, and regulates many physiological processes. Mechanistically, inactivation follows different principles in different ion channels (*Sukomon et al., 2023*). The inactivation rate of PIEZO channels is essential for the physiological functions of different cell types, including neuronal and non-neuronal cells. Moreover, different subtypes and species of PIEZO channels exhibit different inactivation rates. For example, hPIEZO1 and mPIEZO2 have a faster inactivation rate than mPIEZO1 (*Coste et al., 2010*; *Anishkin et al., 2014*; *Bae et al., 2013*). More importantly, abnormal inactivation of the PIEZO channel is one of the dominant outcomes of clinical PIEZO channelopathy (*Coste et al., 2010*; *Bae et al., 2013*). The MDFIC inserts into the PIEZO pore module and significantly reshapes channel inactivation (*Zhou et al., 2023*), which may also link PIEZO channel inactivation to cell fate. The lack of knowledge regarding the faster inactivating hPIEZO1 has prevented the acquisition of structural information about the relationship of true fast inactivating wild-type hPIEZO1 to cell fate determination, as well as clinically significant hPIEZO1 GOF slow-inactivating channelopathies. Our work has the following implications.

First, we present the near-atomic cryo-EM structures of the fast inactivation hPIEZO1 and its slow inactivation channelopathy mutants, illuminating the fast inactivation mechanism of PIEZO channels involved by the pore lipids, which ingeniously seals the hydrophobic pore.

Second, the overall structure of the curved hPIEZO1 shows a more flattened and extended state compared to the curved mPIEZO1, although there is no strong evidence yet for a link between curvature and inactivation. However, based on the mild GOF mutants hPIEZO1-A1988V and hPIEZO1-E756del, which are in the blade arm region, the correlation between curvature and inactivation is relatively evident, as the blade region indeed influences the channel inactivation. And all the force signal will be transduced to the pore region, removing the pore lipid barrier. Therefore, the pore lipid will likely play a key role in PIEZO gating.

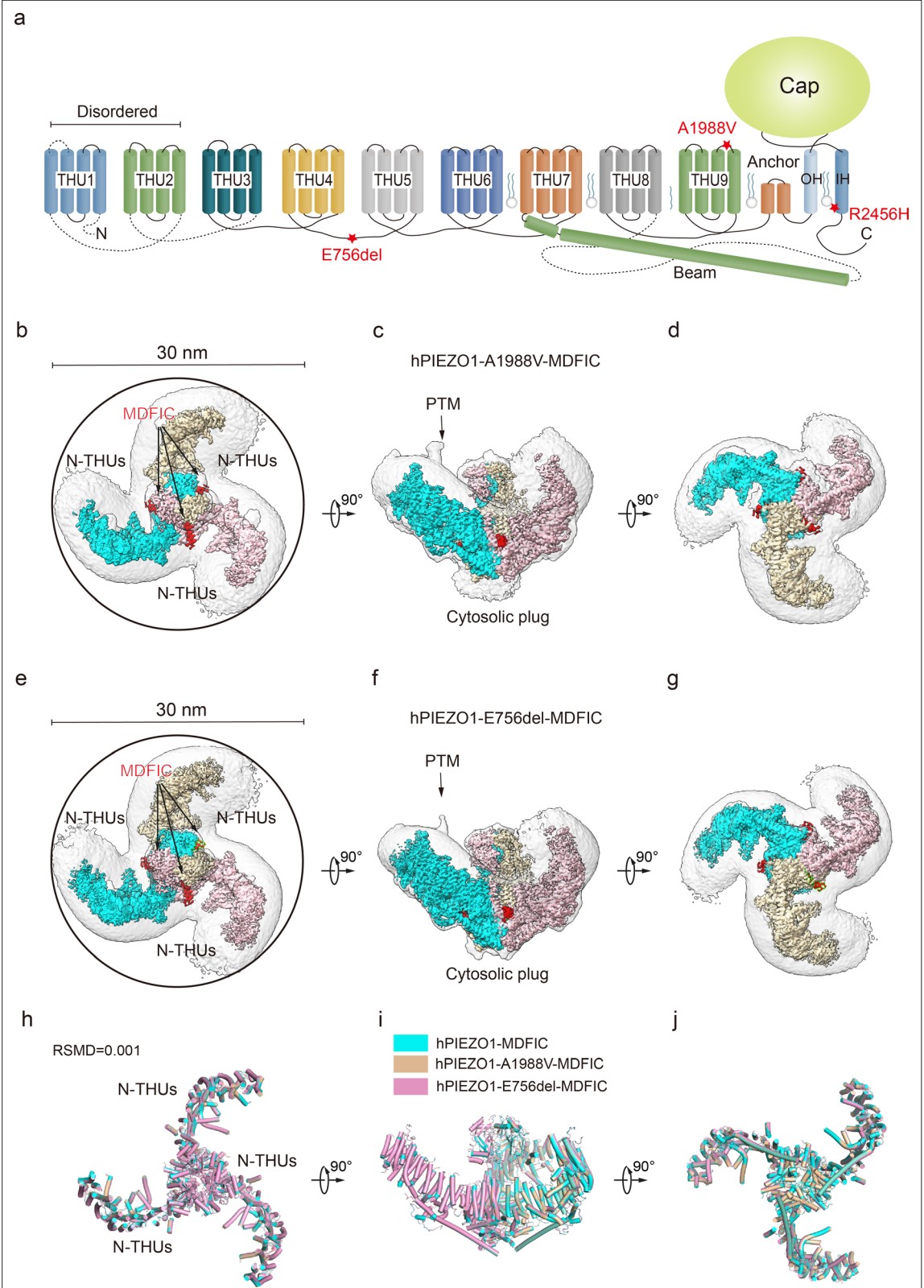

**Figure 4.** Structure of human PIEZO1 (hPIEZO1)-A1988V-MDFIC and hPIEZO1-E756del-MDFIC complex. (**a**) 38-TM topology model of a single hPIEZO1 subunit. The E756del mutation is located at the THU4 and THU5 linker, the A1988V mutation is located at the THU9 linker, and the R2456H mutation is located at the IH of the pore module. (**b**) The 3.1 Å cryo-EM density map of hPIEZO1-A1988V-MDFIC at top view. The density of each single subunit is colored in cyan, wheat, and pink, respectively. The density of the three C-terminal helices of MDFIC is colored in red. Around 30 nm digitonin disk

*Figure 4 continued on next page*

*Figure 4 continued*

is shown as gray density by low pass filtering. (**c**) The 3.1 Å cryo-EM density map of hPIEZO1-A1988V-MDFIC at the side view. The density of each single subunit is colored in cyan, wheat, and pink, respectively. The density of the three C-terminal helices of MDFIC is colored in red. The potential posttranslational modification (PTM) at the transmembrane helix units (THUs) region is indicated. A flexible density binds to the cytosolic plug, which may represent an additional hPIEZO1 auxiliary. (**d**) The 3.1 Å cryo-EM density map of hPIEZO1-A1988V-MDFIC at the bottom view. The density of each single subunit is colored in cyan, wheat, and pink, respectively. (**e**) The 2.8 Å cryo-EM density map of hPIEZO1-E756del-MDFIC at top view. The density of each single subunit is colored in cyan, wheat, and pink, respectively. The density of the three C-terminal helices of MDFIC is colored in red. Around 30 nm digitonin disk is shown as gray density by low pass filtering. (**f**) The 2.8 Å cryo-EM density map of hPIEZO1-E756del-MDFIC at the side view. The density of each single subunit is colored in cyan, wheat, and pink, respectively. The density of the three C-terminal helices of MDFIC is colored in red. The potential PTM at the THUs region is indicated. A flexible density binds to the cytosolic plug, which may represent an additional hPIEZO1 auxiliary. (**g**) The 2.8 Å cryo-EM density map of hPIEZO1-E756del-MDFIC at the bottom view. The density of each single subunit is colored in cyan, wheat, and pink, respectively. (**h**) Superimposed cartoon models of hPIEZO1-MDFIC (cyan), hPIEZO1-A1988V-MDFIC (wheat), and hPIEZO1-E756del-MDFIC (pink) at top view. The RSMD is around 0.001, indicating that they are almost identical. (**i**) Superimposed cartoon models of hPIEZO1-MDFIC (cyan), hPIEZO1-A1988V-MDFIC (wheat), and hPIEZO1-E756del-MDFIC (pink) at the side view. (**j**) Superimposed cartoon models of hPIEZO1-MDFIC (cyan), hPIEZO1-A1988V-MDFIC (wheat), and hPIEZO1-E756del-MDFIC (pink) at bottom view.

The online version of this article includes the following figure supplement(s) for figure 4:

**Figure supplement 1.** Cryo-EM data processing procedure of human PIEZO1 (hPIEZO1)-A1988V mutant.

**Figure supplement 2.** Structural comparison of human PIEZO1 (hPIEZO1) and hPIEZO1-A1988V.

**Figure supplement 3.** Cryo-EM data processing procedure of human PIEZO1 (hPIEZO1)-E756del mutant.

**Figure supplement 4.** Cryo-EM data processing procedure of human PIEZO1 (hPIEZO1)-A1988V co MDFIC.

**Figure supplement 5.** Cryo-EM data processing procedure of human PIEZO1 (hPIEZO1)-E756del co MDFIC.

Lastly, the inactivation can be modulated by extrinsic cues, such as MDFIC, and the channelopathy mutants of PIEZO channels often cause a slower inactivation rate. The primary mechanism of PIEZO channel fast inactivation may provide clues against the clinical mechanopathologies. More importantly, these insights may inspire further investigation into mechanosensing channels as cell fate regulators in the near future.

# Materials and methods

## Constructs

A synthetic codon-optimized gene fragment encoding residues 1–2521 of the hPIEZO1 was cloned into a modified pEG-BacMam vector (*Goehring et al., 2014*) using EcoRI and XhoI restriction enzyme. The resulting protein has enhanced green fluorescent protein (EGFP) and a FLAG-modified antibody recognition sequence (DYKDDDDK) on the C terminus, separated by a PreScission protease (Ppase) cleavage site (LFQ/GP). Other mutations were built on its base by point mutations.

The cDNA for full-length human MDFIC (residues 1–246) was fished from the human cDNA library and cloned into a similarly modified pEG-BacMam vector with no tag.

## Cell lines

*Spodoptera frugiperda* Sf9 cells, Expi-HEK293F suspension cells, and *Escherichia coli* DH10Bac cells were purchased from Thermo Fisher Scientific. Cells were routinely tested for mycoplasma contamination and were negative. Sf9 cells were cultured in Sf-900 II SFM medium (Gibco) at 27°C. Expi-HEK293F cells grown in HEK293 medium (Yocon) at 37°C with 6% $CO_2$.

## Protein expression

hPIEZO1 and its mutations were expressed alone or co-expressed with the MDFIC subunit in expi-HEK293F cells using the BacMam technology. Bacmid carrying hPIEZO1 or MDFIC subunit was generated by transforming *E. coli* DH10Bac cells with the corresponding pEG-BacMam construct, and recombinant bacmids were screened by blue-white spot validation. Baculoviruses were produced by transfecting Sf9 cells at a density of $1 \times 10^6$ per ml with the bacmid using Cellfectin II (Invitrogen). To increase the viral titer, the recombinant virus has undergone two rounds of amplification to generate the P3 virus.

Expi-HEK293F cells in suspension were cultured to a density of $3 \times 10^6$ cells/ml and infected with the P3 virus. For the expression of hPIEZO1 alone, cell culture was infected with 8% (vol:vol) of hPIEZO1

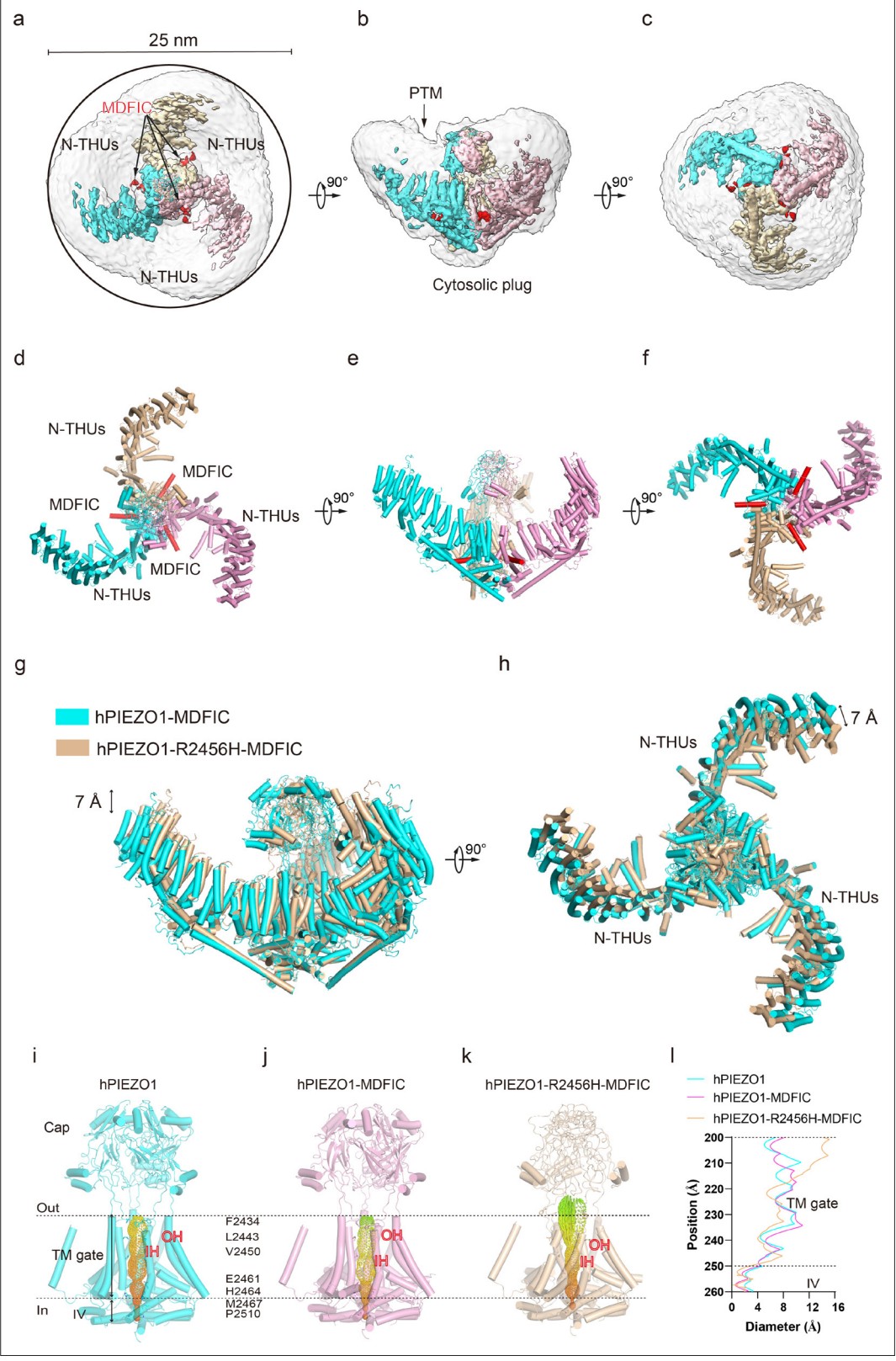

**Figure 5.** Structure of human PIEZO1 (hPIEZO1)-R2456H-MDFIC complex. (**a**) The 4.5 Å cryo-EM density map of hPIEZO1-R2456H-MDFIC at top view. The density of each single subunit is colored in cyan, wheat, and pink, respectively. The density of the three C-terminal helices of MDFIC is colored in red. Around 25 nm digitonin disk of hPIEZO1-R2456H-MDFIC is shown as gray density by low pass filtering, smaller than the wild-type hPIEZO1-

*Figure 5 continued*

MDFIC. (**b**) The 4.5 Å cryo-EM density map of hPIEZO1-R2456H-MDFIC at the side view. The density of each single subunit is colored in cyan, wheat, and pink, respectively. The density of the three C-terminal helices of MDFIC is colored in red. The potential posttranslational modification (PTM) at the transmembrane helix units (THUs) region is indicated. A flexible density binds to the cytosolic plug, which may represent an additional hPIEZO1 auxiliary. (**c**) The 4.5 Å cryo-EM density map of hPIEZO1-R2456H-MDFIC at the bottom view. The density of each single subunit is colored in cyan, wheat, and pink, respectively. The density of the three C-terminal helices of MDFIC is colored in red. (**d**) The cartoon model of hPIEZO1-R2456H-MDFIC at the top view. Each single subunit is colored in cyan, wheat, and pink, respectively. Three C-terminal helices of MDFIC are colored in red. (**e**) The cartoon model of hPIEZO1-R2456H-MDFIC at the side view. Each single subunit is colored in cyan, wheat, and pink, respectively. Three C-terminal helices of MDFIC are colored in red. (**f**) The cartoon model of hPIEZO1-R2456H-MDFIC at the bottom view. Each single subunit is colored in cyan, wheat, and pink, respectively. Three C-terminal helices of MDFIC are colored in red. (**g**) Structural comparison of hPIEZO1-R2456H-MDFIC and hPIEZO1-MDFIC at side view. The motion of the distal blade between hPIEZO1-R2456H-MDFIC and hPIEZO1-MDFIC is around 7 Å from the side view. (**h**) Structural comparison of hPIEZO1-R2456H-MDFIC and hPIEZO1-MDFIC at top view. The motion of the distal blade between hPIEZO1-R2456H-MDFIC and hPIEZO1-MDFIC is around 7 Å from the top view. (**i**) The cartoon model of hPIEZO1 pore module with calculated pore. (**j**) The cartoon model of hPIEZO1-MDFIC pore module with calculated pore. (**k**) The cartoon model of hPIEZO1-R2456H-MDFIC pore module with calculated pore. (**l**) The calculated pore diameter of hPIEZO1 (cyan), hPIEZO1-MDFIC (pink), and flattened hPIEZO1-R2456H-MDFIC (wheat) along the z axis. The hPIEZO1-R2456H-MDFIC presents a more extended extracellular side pore.

The online version of this article includes the following figure supplement(s) for figure 5:

**Figure supplement 1.** Cryo-EM data processing procedure of human PIEZO1 (hPIEZO1)-R2456H mutant.

**Figure supplement 2.** Cryo-EM data processing procedure of human PIEZO1 (hPIEZO1)-R2456H co MDFIC.

**Figure supplement 3.** The EM density of human PIEZO1 (hPIEZO1)-R2456H co MDFIC.

baculovirus. For the co-expression of hPIEZO1 and MDFIC subunit, cell culture was infected with 4% (vol:vol) hPIEZO1 baculoviruses and 4% (vol:vol) MDFIC baculoviruses. After 12 hr, sodium butyrate was added to a 10 mM final concentration, and the temperature was decreased to 30°C. After 60 hr of expression, cells were collected by centrifugation at 4000 r.p.m., 4°C for 10 min, resuspended in Tris-buffered saline (TBS) buffer containing 20 mM Tris pH 7.4, 150 mM NaCl.

## Protein purification

The cell pellet was homogenized in a TBS buffer supplemented with 2 mM phenylmethylsulfonyl fluoride by ultrasonication. Then, large organelles and insoluble matter were pelleted by centrifugation at 8000×$g$ for 10 min. The supernatant was centrifuged at 36,000 r.p.m. for 30 min in a Ti45 rotor (Beckman). The membrane pellet was mechanically homogenized and solubilized in extraction buffer containing TBS, 1% (wt/vol) LMNG, and 0.1% (wt/vol) cholesteryl hemisuccinate for an hour with stirring. Insoluble materials were removed by centrifugation at 36,000 r.p.m. for 1 hr in a Ti45 rotor (Beckman). The supernatant was loaded onto anti-FLAG G1 affinity resin (GenScript) by gravity flow. The resin was further washed with 10 column volumes of wash buffer containing TBS and 0.02% (wt/vol) LMNG, and protein was eluted with an elution buffer containing TBS, 0.02% (wt/vol) LMNG, and 230 µg/ml FLAG peptide. The C-terminal EGFP tag of eluted protein was removed by Ppase cleavage at 4°C overnight. The protein was further concentrated by a 100 kDa cutoff concentrator (Millipore) and loaded onto a Superose 6 increase 10/300 column (GE Healthcare) running in TBS with 0.01% (wt/vol) digitonin. Peak fractions were combined for cryo-EM sample preparation.

## Cryo-EM sample preparation

For cryo-EM sample preparation, 3 µl aliquots of the protein sample were loaded onto glow-discharged (20 s, 15 mA; Pelco easiGlow, Ted Pella) Au grids (Quantifoil, Au R1.2/1.3, 300 mesh). The grids were blotted for 6 s with three forces after waiting for 20 s and immersed in liquid ethane using Vitrobot (Mark IV, Thermo Fisher Scientific) in 100% humidity and 8°C.

## Data collection

Cryo-EM data were collected at a nominal magnification of 215K (resulting in a calibrated pixel size of 0.57 Å) on a Titan Krios (Thermo Fisher Scientific) operating at 300 kV equipped with a K3 or Falcon4i

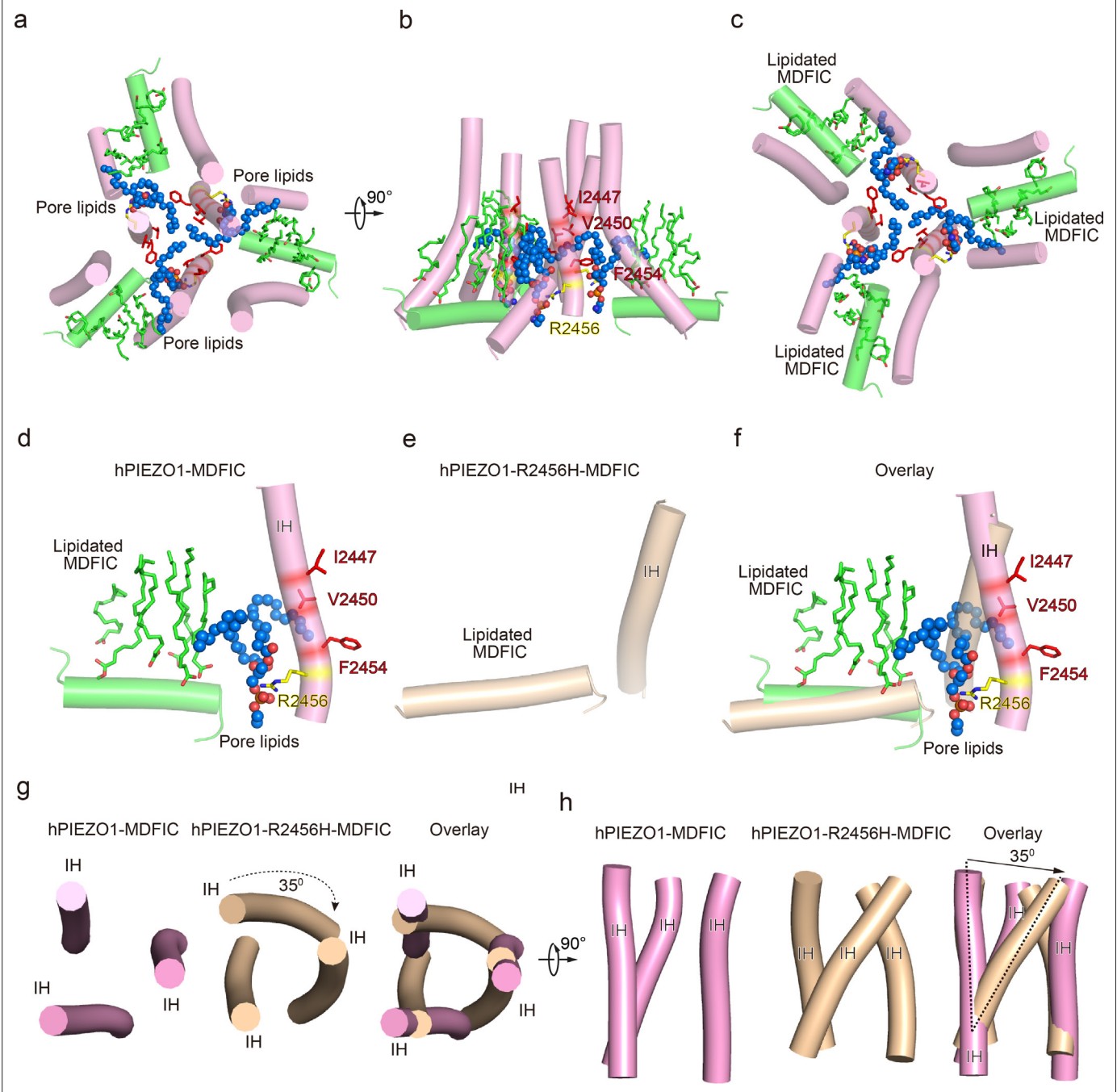

**Figure 6.** The pore module, multi-lipidated MDFIC and pore lipid interaction, and the pore remodeling of the hereditary erythrocytosis (HX) pathogenesis R2456H mutant. (**a**) The cartoon model of human PIEZO1 (hPIEZO1)-MDFIC/hPIEZO1-A1988V-MDFIC/hPIEZO1-E756del-MDFIC pore module at top view. The marine spheres present the pore lipids. (**b**) The cartoon model of hPIEZO1-MDFIC/hPIEZO1-A1988V-MDFIC/hPIEZO1-E756del-MDFIC pore module at side view. The hydrophobic pore region formed by I2447, V2450, and F2454 is marked by red. The R2456 on the lateral side of the inner helix of the pore is marked by yellow. (**c**) The cartoon model of hPIEZO1-MDFIC/hPIEZO1-A1988V-MDFIC/hPIEZO1-E756del-MDFIC pore module at bottom view. The multi-lipidated MDFIC subunits are colored in green and marked. One hydrophobic fatty acid chain of the pore lipid occupies the hydrophobic pore region. Another hydrophobic fatty acid chain of the pore lipid interacts with the fatty acid chains of the MDFIC-covalently linked lipids. In contrast, the hydrophilic head of the pore lipid directly interacts with the R2456 on the lateral side of the inner helix of the pore, thus forming a stable hPIEZO1-multi-lipidated MDFIC-pore lipid complex. The pore lipids seal the pore and prevent ion flow. (**d**) The cartoon model of hPIEZO1-MDFIC/hPIEZO1-A1988V-MDFIC/hPIEZO1-E756del-MDFIC IH, pore lipid, and multi-lipidated MDFIC interactions. The multi-lipidated MDFIC subunits are colored in green. The marine spheres present the pore lipids. The hydrophobic pore region formed by I2447, V2450, and F2454 is marked by red. The R2456 on the lateral side of the inner helix of the pore is marked by yellow. (**e**) The cartoon model of twisted IH and MDFIC

*Figure 6 continued on next page*

*Figure 6 continued*

in hPIEZO1-R2456H-MDFIC. (**f**) The overlay cartoon model of IH and MDFIC in hPIEZO1-MDFIC/hPIEZO1-A1988V-MDFIC/hPIEZO1-E756del-MDFIC and hPIEZO1-R2456H-MDFIC. (**g**) The cartoon model of TM pore in hPIEZO1-MDFIC/hPIEZO1-A1988V-MDFIC/hPIEZO1-E756del-MDFIC and hPIEZO1-R2456H-MDFIC at top view. The TM pore of hPIEZO1-R2456H-MDFIC presents a twisted and extended state. (**h**) The cartoon model of TM pore in hPIEZO1-MDFIC/hPIEZO1-A1988V-MDFIC/hPIEZO1-E756del-MDFIC, and hPIEZO1-R2456H-MDFIC at side view. The TM pore of hPIEZO1-R2456H-MDFIC presents a twisted and extended state.

Summit detector and GIF Quantum energy filter (slit width 20 eV) in super-resolution mode. Movie stacks were automatically acquired using EPU software. The defocus range was set from −0.9 to −1.3 μm. Each movie stack, consisting of 32 frames, was exposed for 2.72 s with a total dose of ~40 e⁻/Å².

## Image processing and model building

Data processing was carried out with cryoSPARC suite (*Punjani et al., 2017*). Patch CTF estimation was carried out after alignment and summary of all 32 frames in each stack using the patch motion

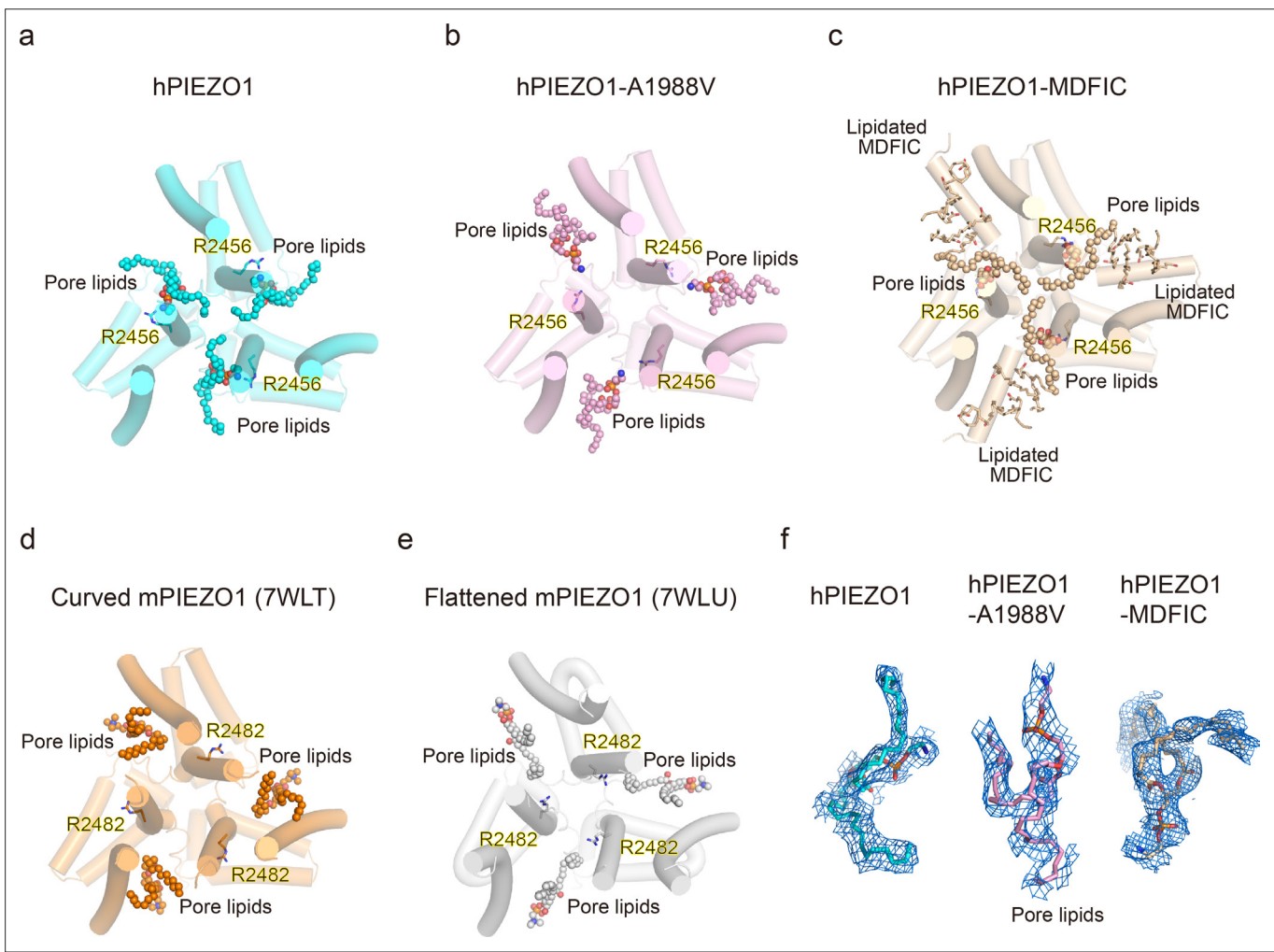

**Figure 7.** Occupancy pattern of pore lipids in human PIEZO1 (hPIEZO1) underlying fast inactivation. (**a**) Top view of hPIEZO1 pore domain. The pore helices are shown as cylinders, and the pore lipid sealing the hydrophobic pore region is shown as spheres. The residue R2456s in inner helices, interacting with pore lipids, is labeled. (**b**) Top view of hPIEZO1-A1988V pore domain. Pore lipids are retracted from the hydrophobic pore region. (**c**) Top view of hPIEZO1-MDFIC pore domain. Lipidated MDFICs are shown as sticks. (**d**) Top view of curved mouse PIEZO1 (mPIEZO1) (PDB: 7WLT). The residues R2482s, responding to R2456 in hPIEZO1, are labeled. Pore lipids are modeled in the latency side of the IH pore. (**e**) Top view of flattened mPIEZO1 (PDB: 7WLU). Pore lipids are also located in the latency side of the IH pore. (**f**) Cryo-EM density of pore lipids in WT hPIEZO1 (cyan), hPIEZO1-A1988V (pink), and hPIEZO1-MDFIC (brown). The cryo-EM density is shown as blue mesh.

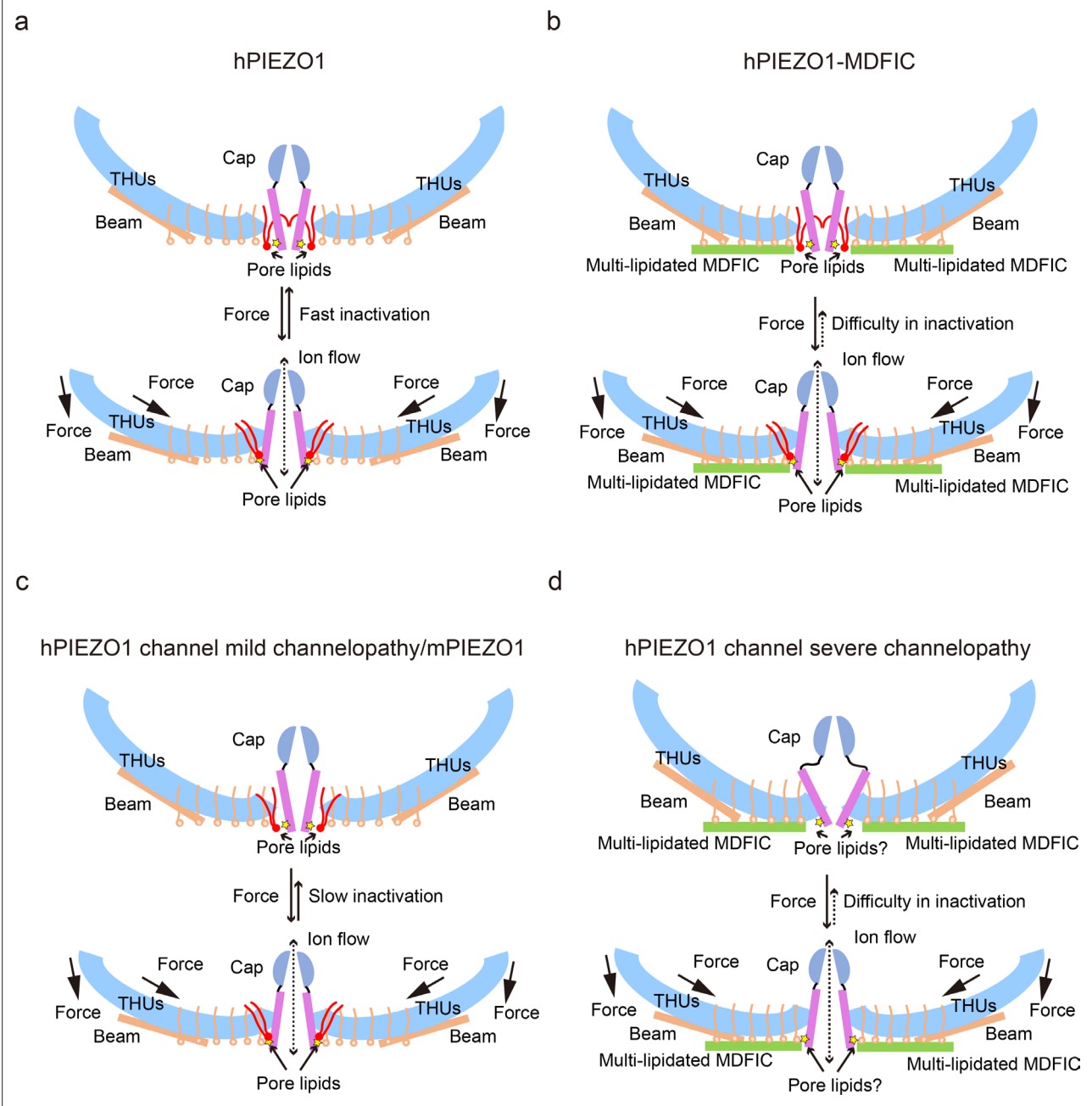

**Figure 8.** The proposed fast inactivation model of PIEZO1 channel. (**a**) Cartoon representation of the fast inactivation model of the human PIEZO1 (hPIEZO1) channel. Two of the three subunits are shown for clarity. The cap (purple semicircle), pore helices (pink rectangle), transmembrane helix units (THUs) (blue arc), beam (brown rectangle), and pore lipids (red) are present. The pore lipids interacting with R2456 (yellow star) seal the TM pore. The membrane force can reduce the membrane curvature and transduce the force to the pore region, removing the barrier created by the pore lipids, which might induce the channel pore to open. The pore lipids then rapidly return to the hydrophobic pore, exhibiting a fast-inactivation pattern. (**b**) Cartoon diagram of the slow inactivation model of the hPIEZO1-MDFIC complex. Multi-lipidated MDFICs are shown as green rectangles. The fatty acid chains of the covalently linked MDFIC lipids can stabilize the pore lipids. Therefore, the hPIEZO1-MDFIC complex exhibits a very stable deep resting state. Once a higher threshold of mechanical force removes the pore lipids, it is also tricky for them to return to the hydrophobic pore region and reform the stable nonconducting pore module, thus exhibiting the prolonged slow inactivation phenotype. (**c**) Cartoon diagram of the slow inactivation model of hPIEZO1 channel mild channelopathy or mouse PIEZO1 (mPIEZO1). Mild channelopathy mutants or mPIEZO1 show the pore lipids retracted from the pore and thus exhibit a slow inactivation pattern. (**d**) Cartoon representation of the slow inactivation model of the hPIEZO1 channel hereditary erythrocytosis (HX) pathogenesis mutant R2456H with MDFIC auxiliary subunit. The R2456H channelopathy mutant probably disrupts the IH pore and the pore lipid interaction. Due to the stabilizing effect of the MDFIC auxiliary subunit, the pore has a twisted shape, and the overall architecture of hPIEZO1-R2456H-MDFIC has a more curved and contracted bowl-like shape.

correction. Initial particles were picked from a few micrographs using blob picker in cryoSPARC, and 2D averages were generated. Final particle picking was done by template picker using templates from those 2D results. After three rounds of 2D classification and 2D selection, ab initio reconstruction, nonuniform refinement, and local refinement, the density map was reconstructed. All maps were low-pass filtered to the map-model FSC value. The reported resolutions were based on the FSC = 0.143 criterion. An initial model was generated by mPIEZO1 (7WLT). Then, we manually completed and refined the model using Coot. Subsequently, the models were refined against the corresponding maps by PHENIX. We used PyMol and UCSF Chimera (*Pettersen et al., 2004*) for structural analysis and graphics generation.

### Electrophysiological recording

PIEZO1-KO-HEK293T cells were a gift obtained from Bailong Xiao Lab and cultured on coverslips placed in a 12-well plate containing DMEM (Gibco) supplemented with 10% fetal bovine serum. The cells in each well were transiently transfected with 1 μg hPIEZO1 or mutant hPIEZO1 plasmids fused with GFP or co-transfected with MDFIC plasmid (weight ratio = 1:3) using polyethyleneimine according to the manufacturer's instructions. After 12–20 hr, the coverslips were transferred to a recording chamber containing the external solution (10 mM HEPES-Na pH 7.4, 150 mM NaCl, 5 mM glucose, 2 mM $MgCl_2$, and 1 mM $CaCl_2$). Borosilicate micropipettes (OD 1.5 mm, ID 0.86 mm, Sutter) were pulled and fire-polished to 2–5 MΩ resistance. For whole-cell recordings, the pipette solution was 10 mM HEPES-Na pH 7.4, 150 mM CsCl, and 5 mM EGTA. The bath solution was 10 mM HEPES-Na pH 7.4, 150 mM NaCl, 5 mM glucose, 2 mM $MgCl_2$, and 1 mM $CaCl_2$.

Recordings were obtained at room temperature (~25°C) using an Axopatch 200B amplifier, a Digidata 1550 digitizer, and pCLAMP 10.7 software (Molecular Devices). The patches were held at −80 mV, and the recordings were low-pass filtered at 1 kHz and sampled at 20 kHz. Mechanical poking was delivered to the cell being patched under whole-cell configuration at an angle of 80° using a fire-polished glass pipette (the tip diameter 3–4 mm). Downward movement of the probe toward the cell was driven by a Clampex-controlled piezoelectric crystal micro-stage (Physik Instrument; E625 LVPZT Controller/Amplifier). The probe had a velocity of 1 μm/ms during the downward/upward motion, and the stimulus was maintained for 1 s. A series of mechanical steps in 1 mm increments was applied every 1.4 s.

### Reporting summary

Further information on research design is available in the Nature Portfolio Reporting Summary linked to this article.

## Acknowledgements

We want to thank the Cryo-EM Facility and High-Performance Computing (HPC) Center of Westlake University for providing cryo-EM and computation support. This work was supported by the National Natural Science Foundation of China (92068201) and Key R&D Program of Zhejiang (2024SSYS0031). We also would like to thank all the Cell Fate Control Lab members for their support.

## Additional information

### Funding

| Funder | Grant reference number | Author |
| --- | --- | --- |
| National Natural Science Foundation of China | 92068201 | Duanqing Pei |
| Key Research and Development Program of Zhejiang Province | 2024SSYS0031 | Duanqing Pei |

The funders had no role in study design, data collection and interpretation, or the decision to submit the work for publication.

## Author contributions

Yuanyue Shan, Data curation, Methodology, Writing - original draft; Xinyi Guo, Data curation, Methodology; Mengmeng Zhang, Meiyu Chen, Ying Li, Data curation; Mingfeng Zhang, Conceptualization, Data curation, Formal analysis, Validation, Methodology, Writing - original draft, Writing - review and editing; Duanqing Pei, Supervision

## Author ORCIDs

Yuanyue Shan http://orcid.org/0000-0001-6291-2433
Mingfeng Zhang https://orcid.org/0000-0002-0138-3934
Duanqing Pei https://orcid.org/0000-0002-5222-014X

Reviewer #1 (Public review): https://doi.org/10.7554/eLife.101923.3.sa1
Reviewer #2 (Public review): https://doi.org/10.7554/eLife.101923.3.sa2
Reviewer #3 (Public review): https://doi.org/10.7554/eLife.101923.3.sa3
Author response https://doi.org/10.7554/eLife.101923.3.sa4

# Additional files

## Supplementary files

MDAR checklist

## Data availability

The cryo-EM reconstructions and final models were deposited with the Electron Microscopy Data Base (accession codes EMD-39205, EMD-60479, EMD-60481, EMD-39219, EMD-65195 and EMD-39223) and with the Protein Data Bank (accession code 8YEZ, 8ZU3, 8ZU8, 8YFC, 9VMX and 8YFG). Source data are provided in this paper.

The following datasets were generated:

| Author(s) | Year | Dataset title | Dataset URL | Database and Identifier |
|---|---|---|---|---|
| Zhang M | 2024 | Human PIEZO1 | http://www.ebi.ac.uk/pdbe/entry/emdb/EMD-39205 | Electron Microscopy Data Bank, EMD-39205 |
| Zhang M | 2024 | Human PIEZO1-A1988V-MDFIC | http://www.ebi.ac.uk/pdbe/entry/emdb/EMD-39219 | Electron Microscopy Data Bank, EMD-39219 |
| Zhang M | 2024 | Human PIEZO1-R2456H_MDFIC | http://www.ebi.ac.uk/pdbe/entry/emdb/EMD-39223 | Electron Microscopy Data Bank, EMD-39223 |
| Zhang M | 2024 | Human PIEZO1-MDFIC | http://www.ebi.ac.uk/pdbe/entry/emdb/EMD-60479 | Electron Microscopy Data Bank, EMD-60479 |
| Zhang M | 2024 | Human PIEZO1-A1988V | http://www.ebi.ac.uk/pdbe/entry/emdb/EMD-60481 | Electron Microscopy Data Bank, EMD-60481 |
| Zhang M | 2024 | Human PIEZO1-E756del-MDFIC | http://www.ebi.ac.uk/pdbe/entry/emdb/EMD-65195 | Electron Microscopy Data Bank, EMD-65195 |
| Zhang M | 2024 | Human PIEZO1 | https://doi.org/10.2210/pdb8yez/pdb | Worldwide Protein Data Bank, 10.2210/pdb8yez/pdb |
| Zhang M | 2024 | Human PIEZO1-A1988V-MDFIC | https://doi.org/10.2210/pdb8yfc/pdb | Worldwide Protein Data Bank, 10.2210/pdb8yfc/pdb |

*Continued on next page*

*Continued*

| Author(s) | Year | Dataset title | Dataset URL | Database and Identifier |
|---|---|---|---|---|
| Zhang M | 2024 | Human PIEZO1-R2456H_MDFIC | https://doi.org/10.2210/pdb8yfg/pdb | Worldwide Protein Data Bank, 10.2210/pdb8yfg/pdb |
| Zhang M | 2024 | Human PIEZO1-MDFIC | https://doi.org/10.2210/pdb8zu3/pdb | Worldwide Protein Data Bank, 10.2210/pdb8zu3/pdb |
| Zhang M | 2024 | Human PIEZO1-A1988V | https://doi.org/10.2210/pdb8zu8/pdb | Worldwide Protein Data Bank, 10.2210/pdb8zu8/pdb |
| Zhang M | 2024 | Human PIEZO1-E756del-MDFIC | https://doi.org/10.2210/pdb9vmx/pdb | Worldwide Protein Data Bank, 10.2210/pdb9vmx/pdb |

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
