## [Editor Report · eLife Assessment]

This is a **useful** revised manuscript that shows a set of data including the first cryo-EM structures of human PIEZO1 as well as structures of disease-related mutants in complex with the regulatory subunit MDFIC, which generate different inactivation phenotypes. The molecular basis of PIEZO channel inactivation is of great interest due to its association with several pathologies. This manuscript provides some structural insights that may help to ultimately build a molecular picture of PIEZO channel inactivation. While the structures are of use and clear conformational differences can be seen in the presence of the auxiliary subunit MDFIC, the strength of the evidence supporting the conclusions of the paper, especially the proposed role for pore lipids in inactivation, is **incomplete**.

---

## [Referee Report · Reviewer #1 (Public review)]

Summary:

This manuscript by Shan, Guo, Zhang, Chen et al., shows a raft of interesting data including the first cryo-EM structures of human PIEZO1. Clearly the molecular basis of PIEZO channel inactivation is of great interest and as such this manuscript provides some valuable extra information that may help to ultimately build a molecular picture of PIEZO channel inactivation. However, the current manuscript though does not provide any compelling evidence for a detailed mechanism of PIEZO inactivation.

Strengths:

This manuscript documents the first cryo-EM structures of human PIEZO1 and gain of function mutants associated with hereditary anaemia. It is also the first evidence showing that PIEZO1 gain of function mutants are also regulated by the auxiliary subunit MDFIC.

Weaknesses:

While the structures are interesting and clear differences can be seen in the presence of the auxiliary subunit MDFIC the major conclusions and central tenets of the paper, especially a role for pore lipids in inactivation, lack data to support them. The post translational modification of PIEZOs auxiliary subunit MDFIC is not modelled as a covalent interaction.

Comments on revisions:

The revisions do absolutely nothing to allay any of the major concerns documented in my initial review of this manuscript.

(1) Mouse vs Human inactivation

Not only is a quantification not provided the literature on this point is still not at all referenced or discussed.

(2) MDFIC -lipidation

Even if they are not assigned in the PDB for illustration they can at least be modelled correctly as covalently bound acyl chains.

(3) Pore lipids and inactivation

None of the explanations are consistent with the data shown.

(4) Cytosolic plug

There is not even any extra discussion provided on this point.

(5) Reduced sensitivity of PIEZO1 in the presence of MDFIC and its regulatory mechanism

No quantification is provided.

(6) Both referencing of the PIEZO1 literature and prose could be improved.

There is little to no attempt to improve the referencing.

---

## [Referee Report · Reviewer #2 (Public review)]

Notably, the authors provide the first structure of human PIEZO1 (hPIEZO1), which will facilitate future studies in the field. They reveal that hPIEZO1 has a more flattened shape than mouse PIEZO1 (mPIEZO1) and has lipids that insert into the hydrophobic pore region. To understand how PIEZO1 GOF mutations might affect this structure and the underlying mechanistic changes, they solve structures of hPIEZO1 as well as two HX causing mild GOF mutations (A1988V and E756del) and a severe GOF mutation (R2456H). Unable to glean too much information due to poor resolution of the mutant channels, the authors also attempt to resolve MCFIC-bound structures of the mutants. These structures show that MDFIC inserts into the pore region of hPIEZO1, similar to its interaction with mPIEZO1, and results in a more curved and contracted state than hPIEZO1 on its own. The authors use these structures to hypothesize that differences in curvature and pore lipid position underlie the differences in inactivation kinetics between wild-type hPIEZO1, hPIEZO1 GOF mutations, and hPIEZO1 in complex with MDFIC.

Strengths:

This is the first human PIEZO1 structure. Thus, these studies become the steppingstone for future investigations to better understand how disease-causing mutations affect channel gating kinetics.

Comments on revisions:

The revised version of the manuscript is stronger and the authors have addressed most of our concerns. The only clarification that remains is data related to the electrophysiology experiments, Figure S2. In the response, the authors mention that they were referring to previously reported mPIEZO1 mutants. However, it is still missing quantification from the human mutant + MDFIC data. This data should be available to the authors and will be more informative than just the representative traces. In the text line 151-152 "Indeed, electrophysiological studies showed that co-expression of these channelopathy mutants with MDFIC resulted in significantly reduced mechanosensitivity and inactivation rate (Fig. S2)." However the updated version does not have any number or the statistics that were performed to indicate significance. I acknowledge that in the response they describe threshold but very descriptively.

---

## [Referee Report · Reviewer #3 (Public review)]

Summary:

In this manuscript, the authors used structural biology approaches to determine the molecular mechanism underlying the inactivation of the PIEZO1 ion channel. To this end, the authors presented structures of human PIEZO1 and its slow-inactivating mutants. The authors also determined the structures of these PIEZO1 constructs in complexes with the auxiliary subunit MDFIC, which substantially slows down PIEZO1 inactivation. From these structures, the authors observed a unique feature of human PIEZO1 in which the lipid molecules plugged the channel pore in fast-inactivating constructs. The authors proposed that these lipid molecules prevent ion permeation and underlie the molecular mechanism of human PIEZO1 inactivation.

Strengths:

Notedly, this manuscript reported the first structures of a human PIEZO1 channel, its channelopathy mutants, and their complexes with MDFIC. The proposed role of pore lipids in modulating PIEZO1 ion permeation is interesting.

Weaknesses:

The authors' conclusion regarding the role of pore lipids in PIEZO inactivation is based on the assumption that all structures of human PIEZO1 resolved in this work represent comparable functional states relevant to channel inactivation. The authors should at least acknowledge that this is a critical assumption that is difficult to validate. The fitting of the lipid molecule to cryo-EM density could be improved.

Comments on revisions:

Upon revision, the authors substantially weakened the statement regarding the correlation between curvature and inactivation. The authors also toned down the statement regarding the role of pore lipids in channel inactivation. However, I have a few additional comments.

(1) As I have stated above, the assumption here is that all structures presented in this work represent comparable functional states relevant to channel inactivation. However, this assumption could be invalid. For example, the WT channel could be in the closed conformation, whereas the mutant could be stabilized in a different functional state. I understand that this is very difficult to test structurally and functionally. Therefore, I think the authors should at least acknowledge this limitation/assumption.

(2) This time, I reviewed the coordinates and the map of the PIEZO1 structures. For example, in the WT channel, the fitting of the lipid to the cryo-EM density is questionable and I personally wouldn't model this lipid in this pose.

---

## [Author Response]

The following is the authors’ response to the original reviews.

**eLife Assessment**
This useful manuscript shows a set of interesting data including the first cryo-EM structures of human PIEZO1 as well as structures of disease-related mutants in complex with the regulatory subunit MDFIC, which generate different inactivation phenotypes. The molecular basis of PIEZO channel inactivation is of great interest due to its association with several pathologies. This manuscript provides some structural insights that may help to ultimately build a molecular picture of PIEZO channel inactivation. While the structures are of use and clear conformational differences can be seen in the presence of the auxiliary subunit MDFIC, the strength of the evidence supporting the conclusions of the paper, especially the proposed role for pore lipids in inactivation, is incomplete and there is a lack of data to support them.

We thank the editors and reviewers for taking the time and effort to review our manuscript. The evidence supporting the key role of pore lipids in hPIEZO1 activation is as follows. i. Compared with wild-type hPIEZO1, the hydrophobic acyl chain tails of the pore lipids retracted from the hydrophobic pore region in slower inactivating mutant hPIEZO1-A1988V (Fig. 7a-b). ii. Previous electrophysiological functional studies revealed that substituting this hydrophobic pore formed by I2447, V2450, and F2454 with a hydrophilic pore prolongs the inactivation time for both PIEZO1 and PIEZO2 channels (PMID: 30628892). iii. In the structure of the HX channelopathy mutant R2456H, the interaction between the hydrophilic phosphate group head of pore lipids and R2456 is disrupted, remodeling the blade and pore module and resulting in a significantly slow-inactivating rate. iv. The interaction between pore lipids and lipidated-MDFIC stabilizes the pore lipids to reseal the pore upon activation of the hPIEZO1-MDFIC complex.

According to previously proposed models for the role of pore lipids in mechanosensitive ion channels, such as MscS (PMID: 33568813), MS K2P (PMID: 25500157) and OSCA channels (PMID: 37402734), the pore lipids seal the channel pores in closed state and could be removed in open state by mechanical force induced membrane deformation, which obeys the force-from-lipids principle. Therefore, in our putative model, the pore lipids seal the hydrophobic pore of hPIEZO1 in the closed state. Upon activation of hPIEZO1, the pore lipids retract from the hydrophobic pore and interact with multi-lipidated MDFIC, stabilizing in the inactivation state. The mild channelopathy mutants make the pore lipids retract from the hydrophobic pore and harder to close upon activation. For the severe channelopathy mutant, the interaction between the pore lipids and R2456 is disrupted, resulting in the missing of pore lipids and significantly slow-inactivating. We fully understand the concern of the role of pore lipids in our proposed model. Therefore, we have toned down our putative model.

**Public Reviews:**

**Reviewer #1 (Public review):**
Summary:This manuscript by Shan, Guo, Zhang, Chen et al., shows a raft of interesting data including the first cryo-EM structures of human PIEZO1. Clearly, the molecular basis of PIEZO channel inactivation is of great interest and as such this manuscript provides some valuable extra information that may help to ultimately build a molecular picture of PIEZO channel inactivation. However, the current manuscript though does not provide any compelling evidence for a detailed mechanism of PIEZO inactivation.Strengths:This manuscript documents the first cryo-EM structures of human PIEZO1 and the gain of function mutants associated with hereditary anaemia. It is also the first evidence showing that PIEZO1 gain of function mutants are also regulated by the auxiliary subunit MDFIC.

We thank reviewer #1 for the encouragement.

Weaknesses:While the structures are interesting and clear differences can be seen in the presence of the auxiliary subunit MDFIC the major conclusions and central tenets of the paper, especially a role for pore lipids in inactivation, lack data to support them. The post-translational modification of PIEZOser# auxiliary subunit MDFIC is not modelled as a covalent interaction.

We fully understand the concern of the role of pore lipids in our proposed model. Therefore, we have toned down our putative model.

The lipids densities of the post-transcriptional modification of PIEZO1 auxiliary subunit MDFIC are shown below. As the lipids densities are not confident, we only use the single-chain lipids to represent them. And the lipidated MDFIC is proven by the MDFIC identification paper.

**Author response image 1. sa4fig1:** 

**Reviewer #2 (Public review):**
Summary:Mechanically activated ion channels PIEZOs have been widely studied for their role in mechanosensory processes like touch sensation and red blood cell volume regulation. PIEZO in vivo roles are further exemplified by the presence of gain-of-function (GOF) or loss-of-function (LOF) mutations in humans that lead to disease pathologies. Hereditary xerocytosis (HX) is one such disease caused due to GOF mutation in Human PIEZO1, which are characterized by their slow inactivation kinetics, the ability of a channel to close in the presence of stimulus. But how these mutations alter PIEZO1 inactivation or even the underlying mechanisms of channel inactivation remains unknown. Recently, MDFIC (myoblast determination family inhibitor proteins) was shown to directly interact with mouse PIEZO1 as an auxiliary subunit to prolong inactivation and alter gating kinetics. Furthermore, while lipids are known to play a role in the inactivation and gating of other mechanosensitive channels, whether this mechanism is conserved in PIEZO1 is unknown. Thus, the structural basis for PIEZO1 inactivation mechanism, and whether lipids play a role in these mechanisms represent important outstanding questions in the field and have strong implications for human health and disease.To get at these questions, Shan et al. use cryogenic electron microscopy (Cryo-EM) to investigate the molecular basis underlying differences in inactivation and gating kinetics of PIEZO1 and human disease-causing PIEZO1 mutations. Notably, the authors provide the first structure of human PIEZO1 (hPIEZO1), which will facilitate future studies in the field. They reveal that hPIEZO1 has a more flattened shape than mouse PIEZO1 (mPIEZO1) and has lipids that insert into the hydrophobic pore region. To understand how PIEZO1 GOF mutations might affect this structure and the underlying mechanistic changes, they solve structures of hPIEZO1 as well as two HXcausing mild GOF mutations (A1988V and E756del) and a severe GOF mutation (R2456H). Unable to glean too much information due to poor resolution of the mutant channels, the authors also attempt to resolve MCFIC-bound structures of the mutants. These structures show that MDFIC inserts into the pore region of hPIEZO1, similar to its interaction with mPIEZO1, and results in a more curved and contracted state than hPIEZO1 on its own. The authors use these structures to hypothesize that differences in curvature and pore lipid position underlie the differences in inactivation kinetics between wild-type hPIEZO1, hPIEZO1 GOF mutations, and hPIEZO1 in complex with MDFIC.Strengths:This is the first human PIEZO1 structure. Thus, these studies become the stepping stone for future investigations to better understand how disease-causing mutations affect channel gating kinetics.

We thank reviewer #2 for the positive comments.

Weaknesses:Many of the hypotheses made in this manuscript are not substantiated with data and are extrapolated from mid-resolution structures.

We fully understand the concern of the role of pore lipids in our proposed model. Therefore, we have toned down our putative model.

**Reviewer #3 (Public review):**
Summary:In this manuscript, the authors used structural biology approaches to determine the molecular mechanism underlying the inactivation of the PIEZO1 ion channel. To this end, the authors presented structures of human PIEZO1 and its slow-inactivating mutants. The authors also determined the structures of these PIEZO1 constructs in complexes with the auxiliary subunit MDFIC, which substantially slows down PIEZO1 inactivation. From these structures, the authors suggested an anti-correlation between the inactivation kinetics and the resting curvature of PIEZO1 in detergent. The authors also observed a unique feature of human PIEZO1 in which the lipid molecules plugged the channel pore. The authors proposed that these lipid molecules could stabilize human PIEZO1 in a prolonged inactivated state.

We thank reviewer #3 for the summary.

Strengths:Notedly, this manuscript reported the first structures of a human PIEZO1 channel, its channelopathy mutants, and their complexes with MDFIC. The evidence that lipid molecules could occupy the channel pore of human PIEZO1 is solid. The authors' proposals to correlate PIEZO1 resting curvature and pore-resident lipid molecules with the inactivation kinetics are novel and interesting.

Thanks for the positive comments.

Weaknesses:However, in my opinion, additional evidence is needed to support the authors' proposals.(1) The authors determined the apo structure of human PIEZO1, which showed a more flattened architecture than that of the mouse PIEZO1. Functionally, the inactivation kinetics of human PIEZO1 is faster than its mouse counterpart. From this observation (and some subsequent observations such as the complex with MDFIC), the authors proposed the anti-correlation between curvature and inactivation kinetics. However, the comparison between human and mouse PIEZO1 structure might not be justified. For example, the human and mouse structures were determined in different detergent environments, and the choice of detergent could influence the resting curvature of the PIEZO structures.

We apologize for the misleading statement about the anti-correlation between curvature and inactivation kinetics of PIEZOs. We cannot conclude that the observation of curvature variation of mPIEZO1 and hPIEZO1 is related to their inactivation kinetics based on structural studies and electrophysiological assay. The difference in structural basis between mPIEZO1 and hPIEZO1 is what we want to state. To avoid this misleading, we have revised the manuscript.

For the concern about detergent, we cannot fully exclude its influence on the curvature of PIEZOs. However, previously reported structures of mPiezo1 (PDB: 7WLT, 5Z10, 6B3R) were in the different detergent environments or in lipid bilayer, but the curvature of mPiezo1 is similar as shown below. Considering the high sequence similarity between mPiezo1 and hPiezo1, we hypothesize that the curvature of both hPiezo1 and mPiezo1 may be unaffected by the detergent.

**Author response image 2. sa4fig2:** Overall structural comparison of curved mPIEZO1 in the lipid bilayer (PDB: 7WLT), mPiezo1 in CHAPS (PDB: 6B3R) and mPiezo1 in Digitonin (PDB: 5Z10).

(2) Related to point (1), the 3.7 Å structure of the A1988V mutant presented by the authors showed a similar curvature as the WT but has a slower inactivating kinetics.

Based on the structural comparison between hPIEZO1 and its A1998V mutant, the retraction of pore lipids from the hydrophobic center pore in hPIEZO1-A1998V is mainly responsible for its slower inactivating kinetics.

(3) Related to point (1), the authors stated that human PIEZO1 might not share the same mechanism as mouse PIEZO1 due to its unique properties. For example, MDFIC only modifies the curvature of human PIEZO1, and lipid molecules were only observed in the pore of the human PIEZO1. Therefore, it may not be justified to draw any conclusions by comparing the structures of PIEZO1 from humans and mice.

Thanks for the constructive suggestion. To avoid this misleading, we have revised the manuscript.

(4) Related to point (1), it is well established that PIEZO1 opening is associated with a flattened structure. If the authors' proposal were true, in which a more flattened structure led to faster inactivation, we would have the following prediction: more opening is associated with faster inactivation. In this case, we would expect a pressure-dependent increase in the inactivation kinetics.

Could the authors provide such evidence, or provide other evidence along this direction?

We appreciate the reviewer’s comment. We are not claiming a relationship between the flattened structure and activation/inactivation. We only present the results of the structure of wild-type/mutant PIEZO1.

(5) In Figure S2, the authors showed representative experiments of the inactivation kinetics of PIEZO1 using whole-cell poking. However, poking experiments have high cell-to-cell variability.

The authors should also show statics of experiments obtained from multiple cells.

We have shown the statics of representative electrophysiology experiments obtained from multiple cells in Figure S2.

(6) In Figure 2 and Figure 5, when the authors show the pore diameter, it could be helpful to also show the side chain densities of the pore lining residues.

We appreciate the reviewer’s suggestion. The side chain of the pore lining restricted residues have been shown in Figure 2 and Figure 5 and the densities of pore domain have been shown in Figure S4 and S14. Interestingly, the pore lining restricted residues in mPIEZO1 and hPIEZO1 is highly conserved.

(7) The authors observed pore-plugging lipids in slow inactivating conditions such as channelopathy mutations or in complex with MDFIC. The authors propose that these lipid molecules stabilize a "deep resting state" of PIEZO1, making it harder to open and harder to inactivate once opened. This will lead to the prediction that the slow-inactivating conditions will lead to a higher activation threshold, such as the mid-point pressure in the activation curve. Is this true?

Yes, it is true. In Figure S2, the MDFIC-induced slow-inactivation conditions in hPIEZO1-MDFIC, hPIEZO1-A1988V-MDFIC, hPIEZO1-E756del-MDFIC and hPIEZO1-R2456H-MDFIC result in larger half-activation thresholds than hPIEZO1, hPIEZO1-A1988V, hPIEZO1-E756del and hPIEZO1-R2456H, respectively.

**Recommendations for the authors:**

**Reviewer #1 (Recommendations for the authors):**
I document the major issues below:(1) Mouse vs Human inactivationLine 21- "than the slower inactivating curved mouse PIEZO1 (mPIEZO1)."Where is the data in this paper or any other paper that human PIEZO1 inactivates faster than mouse PIEZO1? This is central to the way the authors present the paper. In fact, the tau quoted for the hPIEZO1 of ~10 ms is similar to that often measured for mPIEZO1. The reference in the discussion for mouse vs human inactivation times is a review of mechanotransduction. Either the authors need to directly compare the tau of mP1 vs hP1 or quote the relevant primary literature if it exists.

As measured in HEK-PIKO cells transfected with mPiezo1, the inactivation time of mPiezo1 is 13 ± 1 ms (PMID: 29261642) at -80 mV.

The tau is also voltage-dependent. The tau is beyond 20 ms at -60 mV for mPIEZO1 PMID:

1. and for hPIEZO1 is still around 10 ms.

(2) MDFIC-lipidationWithout seeing the PDB or EMDB I can't guarantee this but from Figure 6d it seems like the Sacylation in the distal C-terminus of MDFIC is not modelled as a covalent interaction, these lipids are covalently added to the Cys residues in S-acylation via zDHHC enzymes. This should be modelled correctly.

Thanks for this suggestion. As the lipid densities of the post-transcriptional modification of PIEZOs auxiliary subunit MDFIC are not confident, we only use the single-chain lipids to represent them.

And the lipidated MDFIC is proven by the MDFIC identification paper (PMID: 37590348).

(3) Pore lipids and inactivationThe lipids close to the pore are interesting and the density for a lipid is also seen in the mouse MDFIC-PIEZO1 complex from Zhou, Ma et al, 2023. However, there is no data provided by the authors that the lipid is functionally relevant to anything. There is not even a correlation with inactivation in Figure 7. P1+MDFIC inactivates slowest yet the lipids are present within the pore. Second, there is no evidence for what these structures are: closed, or inactivated? In fact, the Xiao lab is now interpreting the 7WLU structure as inactivated.

The evidence supporting the key role of pore lipids in hPIEZO1 activation is as follows. i. Compared with wild-type hPIEZO1, the hydrophobic acyl chain tails of the pore lipids retracted from the hydrophobic pore region in slower inactivating mutant hPIEZO1-A1988V (Fig. 7a-b). ii. Previous electrophysiological functional studies revealed that substituting this hydrophobic pore formed by I2447, V2450, and F2454 with a hydrophilic pore prolongs the inactivation time for both PIEZO1 and PIEZO2 channels (PMID: 30628892). iii. In the structure of the HX channelopathy mutant R2456H, the interaction between the hydrophilic phosphate group head of pore lipids and R2456 is disrupted, remodeling the blade and pore module and resulting in a significantly slow-inactivating rate. iv. The interaction between pore lipids and lipidated-MDFIC stabilizes the pore lipids to reseal the pore upon activation of the hPIEZO1-MDFIC complex. Overall, the pore lipid is involved in inactivation, and we have toned down the statement.

(4) Cytosolic plugThere is additional cytosolic density for the human PIEZO1 that the authors intimate could be from a different binding partner. IS it possible to refine this density? Is it from the PIEZO1-tag? At the very least a little more information about this density should be given if it is going to be mentioned like this.

Our purification result shows that the protein is tag-free. We are also curious about the extra cytosolic density, but we do not know what it is.

(5) Reduced sensitivity of PIEZO1 in the presence of MDFIC and its regulatory mechanismThis was reported in the first article however no data is presented by the authors to support MDFIC increasing the mechanical energy required to open PIEZO1. The sentence in the discussion; "MDFIC enables hPIEZO1 to respond to different forces by modifying the pore module through lipid interactions." is not supported by any functional data and seems to be an over-interpretation of the structures.

We appreciate this suggestion. The half-activation threshold of hPEIZO1 and hPEIZO1-MDFIC is measured to be 7 μm and 9 μm, respectively (Fig.S2). In addition, the mechanical currents amplitude of hPIEZO1-MDFIC is extremely small compared to that of WT reaching the nA level (Fig.S2). Therefore, the less mechanosensitive hPIEZO1-MDFIC may require more mechanical energy to open than PIEZO1 WT.

1. Both referencing of the PIEZO1 literature and prose could be improved.

Thanks for the suggestion. We have improved the referencing and prose.

**Reviewer #2 (Recommendations for the authors):**
(1) The authors speculate that the difference in curvature between human and mouse PIEZO1 results in its fast inactivation but do not provide experimental evidence to support this idea. This claim would have been bolstered by showing that the GOF human mutations have a more curved structure, but these proved too structurally unstable to be solved at high resolution. However, the authors state that the 3.7 angstrom map solved for hPIEZO1-A1988V does have an overall similar architecture as wild-type hPIEZO1; thus, contradicting their hypothesis.

We apologize for the misleading statement. In our revised manuscript, we do not claim a relationship between the flattened structure and activation/inactivation. We only present the results of the structure of wild-type/mutant PIEZO1.

The structure comparison between the A1988V mutant and WT shows a similar architecture but a different occupancy pattern of pore lipids. Therefore, we suggested that the A1988V mutant has slightly slower inactivation kinetics, mainly due to the exit of pore lipids from the pore.

(2) The authors show that interaction with MDFIC alters hPIEZO1 structure to be more curved and use this to support their idea that changing the curvature of the protein underlies the prolonged inactivation kinetics. It has been previously shown that MDFIC does not change the structure of mPIEZO1 but does alter its inactivation and gating kinetics. How does this discrepancy fit into the inactivation model proposed by the authors? Similarly, their claim that MDFIC slows hPIEZO1 inactivation and weakens mechanosensitivity just by affecting the pore module and changing blade curvature is made based on observation and no experimental data to test it.

We have revised the manuscript to avoid misleading the relationship between the curvature and the inaction kinetics of hPIEZO1. The evidence reported previously that substitution of the hydrophobic pore, formed by I2447, V2450, and F2454, with a hydrophilic pore prolongs the inactivation time for both PIEZO1 and PIEZO2 channels (PMID: 30628892). In addition, the severe HX channelopathy mutant R2456H, wherein the interaction between the hydrophilic phosphate group head and R2456 is disrupted, leads to remodeling of the blade and pore module. Indeed, our observation is limited and further experiments will be performed to support our model.

(3) How does their model fit in cell types that have PIEZO1 (or GOF mutant PIEZO1) but not MDFIC?

In cell types that have PIEZO1 or GOF mutant PIEZO1 but not MDFIC, PIEZO1 or GOF mutant PIEZO1 may have a faster inactivation rate than those that bind to MDFIC. It can be proved that overexpressed PIEZOs exhibit faster inactivation kinetics than those in some native cell types with MDFIC expression (PMID: 20813920, 30132757).

(4) Figure S2 is missing quantification of the electrophysiology data. The authors should show summary data in addition to their representative traces including the Imax for all conditions, tau for data shown in b, and sample size for all conditions, and related statistics. The text claims that MDFIC decreases mechanosensitivity (line 156) but there is no data to support this.

For the electrophysiological assay in Figure S2, we referred to previously reported mPIEZO1 mutants (PMID: 23487776, 28716860). We confirmed that the slower inactivation phenotypes of these mutations of hPIEZO1 are similar to those of mPIEZO1.

The half-activation threshold of hPEIZO1 and hPEIZO1-MDFIC is measured to be 7 μm and 9 μm, respectively. This tendency of increased half-activation threshold of hPIEZO1 upon binding with MDFIC is also shown in the electrophysiological result of hPIEZO1 channelopathy mutants.

(5) In line 144, the authors mention that they were able to validate the MDFIC density with multilipidated cysteines on the C-terminal amphipathic helix, but they do not show the density with fitted lipids. While individual densities for some of the lipids are shown in extended Figure 12, it would be helpful to include a figure where they show the map for MDFIC with fitted lipids in it.

Thanks for the valuable suggestion. As the lipid densities of the post-transcriptional modification of PIEZOs auxiliary subunit MDFIC are not confident, we only use the single-chain lipids to represent them. And the lipidated MDFIC is proven by the MDFIC identification paper.

(6) The authors show that R2456 interacts with a lipid at the pore module and hypothesize that this underlies the fast inactivation of hPIEZO1. While they did not obtain a high-resolution structure of this mutant, this hypothesis could be tested by substituting R for side chains with different charges and performing electrophysiology to determine the effects on inactivation.

Thanks for the constructive suggestion. We will perform the electrophysiology assay for R2456 mutants with different side chains.

1. Figure 4 shows overall structure of hPIEZO1 GOF mutations A1988V and E756del in complex with MDFIC. Other than showing an overall similar structure to wildtype hPIEZO1, the authors do not show how the human mutations A1988V alter the structure of the protein at the site of change. Understanding how these mutations affect the local architecture of the protein has important relevance for human physiology.

As the GOF channelopathy mutant hPIEZO1-A1988V is structurally unstable, the density at the site of A1988V is too weak to figure out the related interaction in the structure of the hPIEZO1-A1988V mutant.

Minor comment:In general, the manuscript will benefit from heavy copy editing. For example, the word cartoon is misspelled in many of the figure legends.

We apologize for the mistake. The manuscript has been checked and revised.

**Reviewer #3 (Recommendations for the authors):**
Some portions of this manuscript were not well written. For example, at the end of the 3rd paragraph in the introduction, the authors talked about HX mutations and their correlation with malaria infection and plasma iron. This is irrelevant information and will only distract the readers. It would be ideal if the authors could go through the entire manuscript and improve its clarity.

Thanks for the suggestion. We have revised the sentences about HX mutations as suggested and improved the entire manuscript.